# Mechanisms of inhibition and activation of extrasynaptic αβ GABAₐ receptors

Vikram Babu Kasaragod[1,2], Martin Mortensen[3], Steven W. Hardwick[4], Ayla A. Wahid[1], Valentina Dorovykh[3], Dimitri Y. Chirgadze[4], Trevor G. Smart[3✉] & Paul S. Miller[1✉]

Type A GABA (γ-aminobutyric acid) receptors represent a diverse population in the mammalian brain, forming pentamers from combinations of α-, β-, γ-, δ-, ε-, ρ-, θ- and π-subunits[1]. αβ, α4βδ, α6βδ and α5βγ receptors favour extrasynaptic localization, and mediate an essential persistent (tonic) inhibitory conductance in many regions of the mammalian brain[1,2]. Mutations of these receptors in humans are linked to epilepsy and insomnia[3,4]. Altered extrasynaptic receptor function is implicated in insomnia, stroke and Angelman and Fragile X syndromes[1,5], and drugs targeting these receptors are used to treat postpartum depression[6]. Tonic GABAergic responses are moderated to avoid excessive suppression of neuronal communication, and can exhibit high sensitivity to $Zn^{2+}$ blockade, in contrast to synapse-preferring α1βγ, α2βγ and α3βγ receptor responses[5,7–12]. Here, to resolve these distinctive features, we determined structures of the predominantly extrasynaptic αβ GABAₐ receptor class. An inhibited state bound by both the lethal paralysing agent α-cobratoxin[13] and $Zn^{2+}$ was used in comparisons with GABA–$Zn^{2+}$ and GABA-bound structures. $Zn^{2+}$ nullifies the GABA response by non-competitively plugging the extracellular end of the pore to block chloride conductance. In the absence of $Zn^{2+}$, the GABA signalling response initially follows the canonical route until it reaches the pore. In contrast to synaptic GABAₐ receptors, expansion of the midway pore activation gate is limited and it remains closed, reflecting the intrinsic low efficacy that characterizes the extrasynaptic receptor. Overall, this study explains distinct traits adopted by αβ receptors that adapt them to a role in tonic signalling.

Type A GABA (GABAₐ) receptors belong to the pentameric ligand-gated ion channel (pLGIC) superfamily, which includes mammalian nicotinic acetylcholine receptors (nAChRs), serotonin type 3A receptors and glycine receptors, as well as other non-mammalian homologues[14,15]. GABAₐ αβ receptors share common properties regardless of specific α or β subtype[16], comprise a notable population of extrasynaptic receptors[8,17–19], and are important model receptors for understanding drug modulation[20,21]. They bear two distinct traits common to tonic GABAergic conductance. The first is a low open-channel probability ($P_o$) in response to GABA[7,8,21] that avoids over-damping neuronal circuitry. The second is a high sensitivity[9] to inhibition by endogenous $Zn^{2+}$, with αβ receptors being the most sensitive of all isoforms, which has physiological and pathological consequences during development and in conditions such as temporal lobe epilepsy[22,23]. Here we solve structures of extrasynaptic GABAₐ receptors to explain the molecular mechanisms underlying a low $P_o$ and marked inhibition by $Zn^{2+}$.

We reconstituted purified α1β3 receptor pentamers into lipid nanodiscs[24] and solved structures of the receptor in complex with α-cobratoxin (α-CBTx)–$Zn^{2+}$ (to 3.0 Å resolution), GABA–$Zn^{2+}$ (2.8 Å) or GABA (3.0 Å) (Extended Data Fig. 1, Extended Data Table 1). Looking down from the extracellular side onto the pentamer, the subunit order

and stoichiometry read α–β–β–α–β in a clockwise direction, such that the third subunit (β (chain C), underlined) occupies the equivalent of the 'γ-position' in synaptic αβγ receptors[25–27] (Fig. 1a). In all the structures, the single β–β interface is occupied by megabody 25 (Mb25), which comprises the immunogenic binding domain nanobody 25 (Nb25) fused to a cHopQ enlargement domain, which is required to randomize particle orientation and break the quasi-five-fold symmetry for particle alignment[25,28]. The co-ligand histamine, included to boost yield, also occupies this β–β interface in all the structures, binding deeper inside the crevice in a pocket homologous to the two orthosteric β–α GABA binding sites, as previously described[28,29]. Whole-cell patch-clamp recording confirmed that the construct that we imaged by cryo-electron microscopy (cryo-EM) (Methods) exhibited the sensitivity, desensitization, current response size and weak histamine modulation[30] of the wild-type receptor (Extended Data Fig. 2). Nb25 exerted a weak positive allosteric modulation at the highest concentration tested (10 μM) that was not observed with α1β3γ2 receptors, which lack the β–β interface (Extended Data Fig. 3). In contrast to αβγ receptors, the N-linked glycans at Asn111 in α1 are not resolved by electron density inside the vestibule (Fig. 1a), consistent with the absence of the γ2-subunit Trp123 (β3 chain C Gly108 in αβ receptors), which stacks under the α1 chain A glycan to impose order[25,27].

[1]Department of Pharmacology, University of Cambridge, Cambridge, UK. [2]MRC Laboratory of Molecular Biology, Cambridge, UK. [3]Department of Neuroscience, Physiology and Pharmacology, University College London, London, UK. [4]Cryo-EM Facility, Department of Biochemistry, University of Cambridge, Cambridge, UK. ✉e-mail: t.smart@ucl.ac.uk; pm676@cam.ac.uk

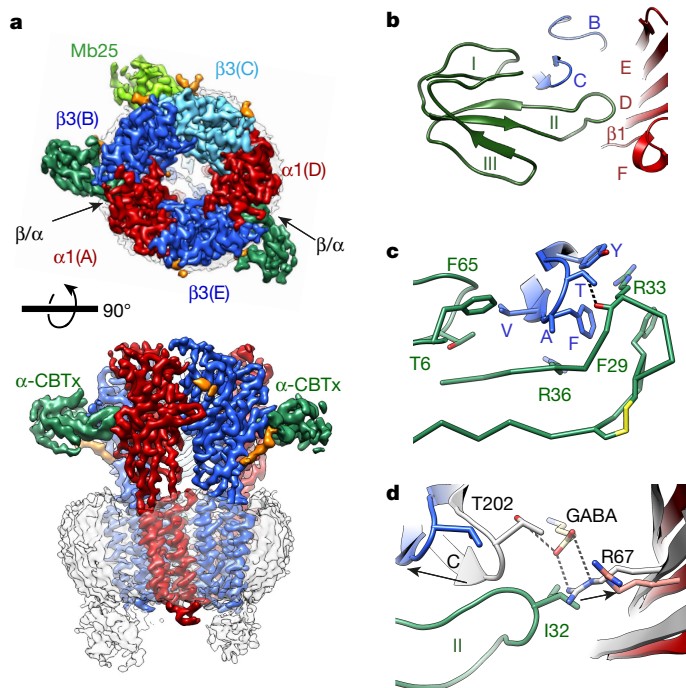

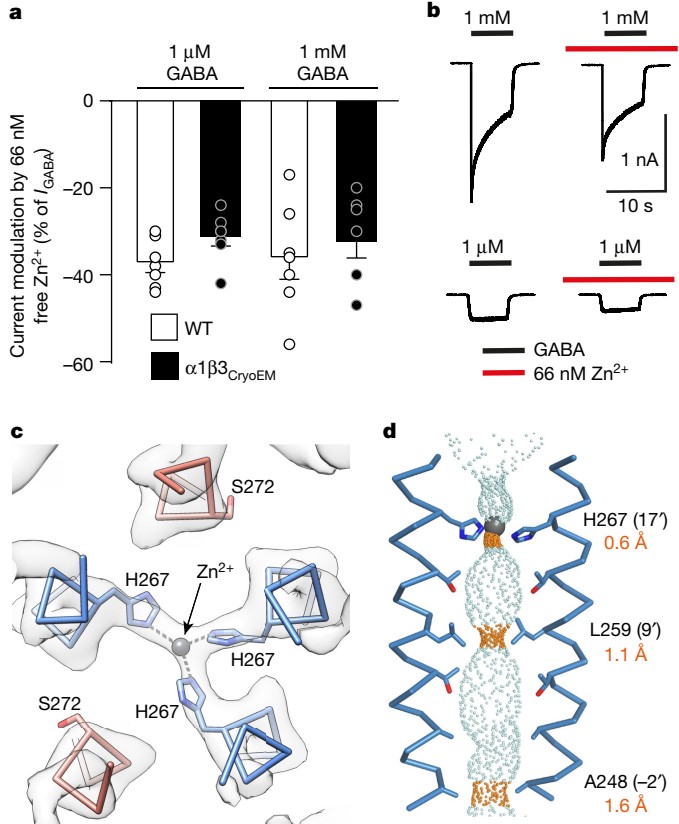

**Fig. 1 | α-Cobratoxin binding site on α1β3 GABA$_A$ receptors. a**, α-CBTx–Zn$^{2+}$-bound α1β3 receptor cryo-EM map, showing top (top) and side (bottom) views. α-CBTx is bound to the β–α neurotransmitter pocket interface. Glycans (orange) are not resolved inside the vestibule (top). Mb25 is shown in lime green; nanodisc and 'hanging' β3-subunit thermostabilized apocytochrome b562RIL (BRIL) densities are in grey. **b**, Atomic model of α-CBTx (green) bound to the GABA$_A$ receptor with finger II positioned at the β3 (blue)–α1 (red) interface. **c**, Close up of the binding mode in **b** showing residue positions and interactions (β3 loop C residues in blue are Val199, Phe200, Ala201, Thr202 and Tyr205). **d**, Overlays of the GABA-bound model (white) and α-CBTx-bound model (β3 loop C in blue, α1 with Arg67 in pink and red, and α-CBTx finger II in green), showing that finger II does not directly overlap with the GABA binding pose but displaces loop C and Arg67 away (black arrows) so that they no longer support GABA binding. Dashed lines represent putative hydrogen-bond interactions.

## Mechanism of inhibition by α-CBTx

α-Cobratoxin (α-CBTx) blocks muscle nAChRs to paralyse prey, but has also been shown to act with reduced potency as an inhibitor of GABA$_A$ receptors in recombinant expression systems[13]. Such toxins represent new scaffolds for subtype-selective inhibitor design, but to our knowledge, there are no available structures of GABA$_A$ receptors in complex with protein inhibitors[13,31,32] to reveal modes of action and guide rational engineering approaches[33]. α-CBTx bridges the β–α interface of the receptor halfway up the outer extracellular domain (ECD) at both GABA binding pockets (Fig. 1a). The characteristic three β-strand loops ('fingers') I–III of α-CBTx dock perpendicular to the cylindrical GABA$_A$ receptor to encase loop C, an essential responsive element to neurotransmitter binding[26,27,34,35] (Fig. 1a–c, Extended Data Fig. 4a, b). Thr6 and Phe65 of α-CBTx form van der Waals interactions with the receptor loop C Val199. Finger II inserts into the neurotransmitter pocket (between loops β3 B and C, and the α1 loop D, E and F β1-strands), forming the key contact zone below loop C. The positively charged side chains of Arg33 and Arg36 straddle the aromatic side chain of loop C Phe200. Arg33 also stacks below the Tyr205 aromatic side chain, and its backbone carbonyl contributes a putative hydrogen bond with the Thr202 hydroxyl 2.7 Å away. The binding mode resembles the one solved at 4.2 Å resolution for α-CBTx bound to the AChBP from *Lymnaea stagnalis*, a soluble homologue of nAChR[36], and for α-bungarotoxin-bound muscle nAChR[37] (Extended Data Fig. 4c–h).

**Fig. 2 | The Zn$^{2+}$ binding site. a**, Bar chart showing inhibition of maximal (1 mM) and EC$_{20}$ (1 μM) GABA whole-cell current responses ($I_{GABA}$) by 66 nM free Zn$^{2+}$ (controlled using the chelator tricine) for wild-type αβ (WT) and the α1β3 cryo-EM construct (α1β3$_{CryoEM}$) expressed in HEK 293 cells. Data are mean ± s.e.m. $n = 7$ for wild-type and cryo-EM constructs, from biologically independent patch-clamp experiments with individual cells. One-way analysis of variance (ANOVA) and Tukey multiple comparisons post hoc test showed no significant differences across groups, $F_{(3, 24)} = 0.6449$; $P = 0.5937$. **b**, Corresponding current recordings for α1β3$_{CryoEM}$. **c**, Top view of C$_α$ backbones of M2 pore-lining helices showing three 17′ β3 His267 (blue) residues coordinating Zn$^{2+}$ across the pore (α1 17′ Ser272 residues in red). Cryo-EM map shown as white transparent. **d**, Side-on view of β3-subunit chain B and E M2 helices flanking the pore permeation pathway (blue dots) with narrowings (orange dots) for three closed 'gates' at the 17′ Zn$^{2+}$ site, 9′ hydrophobic (activation) gate and −2′ intracellular (desensitization) gate to create a triple-gated closed pore.

Consistent with conserved roles in binding, Arg33Gly and Arg36Gly mutations reduce affinity by 300-fold at nAChRα7[33]. The GABA$_A$ receptor α-subunits lack the apex loop C aromatic residue (Phe200 in β3), explaining why α-CBTx does not bind at its α–β interfaces. This suggests that α-CBTx will also not bind α–γ or γ–β interfaces in αβγ receptors, although it might bind δ- and ρ-subunit loop C, which also possess this aromatic variant.

α-CBTx does not overlap with and directly antagonize GABA binding (Fig. 1d). Instead, α-CBTx induces rearrangement of the α-subunit Arg67 and an outward motion of the β-subunit loop C by 5.9 Å, thus perturbing two crucial components of the GABA binding site[26,27] (Fig. 1d). The equivalent residue to α1 Ser69—the neighbouring residue to Arg67—is a Lys in α2, and this reduces sensitivity to α-CBTx fivefold[13]. This position is too distal to interfere directly with toxin binding, but the structure reveals that the lysine could exert a steric and electrostatic repulsion on Arg67 that hinders its reorganization to accommodate α-CBTx finger II (Extended Data Fig. 4i, j).

The outward motion of loop C, which is usually associated with antagonist binding to pLGICs, is larger than the one caused by

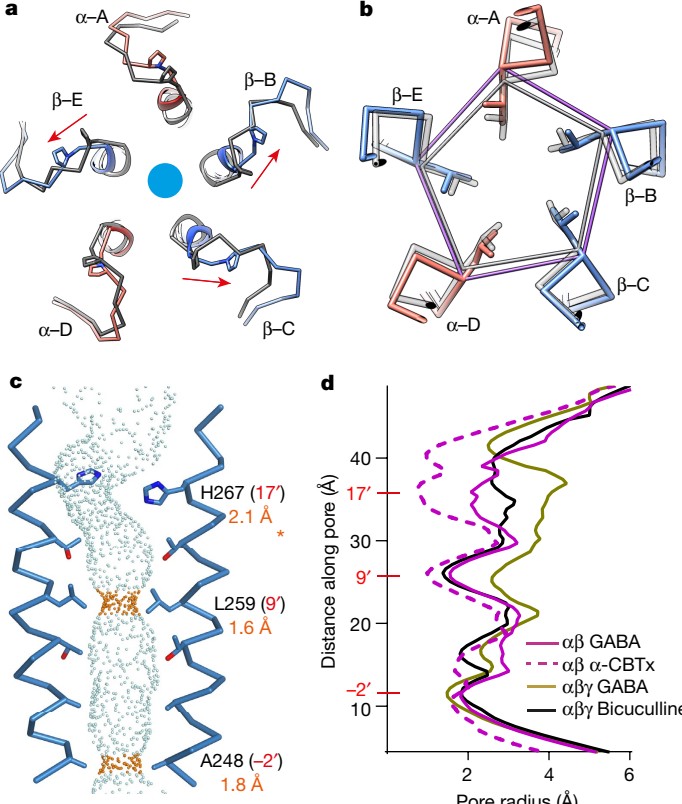

**Fig. 3 | Response of the TMD to GABA binding. a,** Top-down views of α-CBTx–Zn²⁺ (grey) and GABA-bound (red and blue) atomic model overlays showing the M2 helices, M2–M3 linkers, α1 Pro278 and β3 Pro273. The GABA binding β3 B–E subunit linkers respond and switch to the 'outward' conformation (arrows). **b,** Cross-section at the 9′ Leu ring showing expanded Cα pentagonal perimeter for GABA (purple) compared with α-CBTx–Zn²⁺ (grey). **c,** Side-on view

of the permeation pathway (blue and orange dots) between opposing β3-subunit chain B–E M2 helices, showing closed 9′ and −2′ hydrophobic gates. The asterisk indicates the kink in the permeation pathway around the 17′ residue, which varies depending on mobile His side chain positioning, so the 17′ radius of 2.1 Å is indicative only. **d,** Pore radius along the permeation pathway.

the competitive antagonist bicuculline[26] in αβγ receptors (2.1 Å) (Extended Data Fig. 4k). Globally however, the ECD conformation is similar to the bicuculline-bound inhibited state of the αβγ receptor (ECD Cα root mean square deviation (r.m.s.d.) = 1.0 Å, β-subunit chains B and E r.m.s.d. = 0.8 Å), rather than the αβγ GABA-bound state, which features realigned GABA-binding β-subunits[26] (ECD Cα r.m.s.d. = 1.5 Å, β-subunit chains B and E r.m.s.d. = 1.8 Å) (Extended Data Fig. 4l–n). Thus, with respect to receptor conformation, α-CBTx mimics a small competitive antagonist to stabilize an inhibited state of the receptor.

## Zn²⁺ mechanism of channel blockade

The divalent transition metal cation Zn²⁺ is a non-competitive inhibitor of αβ and αβγ GABA_A receptors[17,38,39]. We reproduced this effect in whole-cell patch-clamp recordings, showing that 66 nM free Zn²⁺ inhibited submaximal (20%) (EC₂₀) 1 μM and maximal 1 mM GABA responses by the same amount: 38 ± 2% and 36 ± 6%, respectively, for the wild-type receptor; and 32 ± 2% and 32 ± 4% for the α1β3 cryo-EM construct, (Fig. 2a, b). Sensitivity to inhibition by free Zn²⁺ was the same for the wild-type receptor and the α1β3 cryo-EM contruct (Extended Data Fig. 5a, b). We observed non-protein density that could accommodate a coordinated Zn²⁺ ion at the extracellular end of the pore in the GABA–Zn²⁺–receptor cryo-EM map (2.79 Å resolution), which was absent in the GABA–receptor map (3.04 Å resolution) (Extended Data Fig. 5c–e). No other densities attributable to Zn²⁺ were observed.

The Zn²⁺ site comprises a triad of His267 side chains from the pore-lining M2 helices of the three β3 subunits (Fig. 2c). The τ (far)

nitrogen of each imidazole ring is positioned approximately equidistant, 2.5–2.9 Å, from the Zn²⁺ ion, and at approximately 120°, despite the pseudo-five-fold symmetry of the pore. This location is consistent with its non-competitive mode of antagonism and voltage dependence[38]. Alanine substitution of His267 ablates the high sensitivity to Zn²⁺ inhibition[9,40]. Replacement of one β-subunit His with Ser (as in the δ-subunit) or Ile (as in the γ-subunit) in αβδ and αβγ receptors reduces Zn²⁺ sensitivity 50-fold and 200-fold respectively, explaining the basis of the exquisite subtype selectivity[9]. The triad of His side chains resembles dynamic 'catalytic' sites rather than obligate-bound 'structural' sites involving four sulfur-containing residues[41]. In catalytic sites, an activated water molecule usually completes the tetrahedral coordination[41], but we could not visualize an ordered water molecule in the 2.8 Å-resolution map of the hydrated pore.

Previous GABA_A receptor structures have shown how blockade is achieved at the intracellular end of the pore, for example, by picrotoxinin or cations[26,37,42]. Our structure reveals how blockade operates at the extracellular end of a pLGIC. Zn²⁺ binds a channel conformation with a closed midway hydrophobic gate (9′ leucine ring) and intracellular (−2′ ring) gate, as previously described for αβγ receptors[26,37] (Fig. 2d). However, the Zn²⁺ site creates an additional top gate to prevent the passage of chloride ions from the vestibule into the channel (Fig. 2d). By contrast, the GABA structure shows that in the absence of Zn²⁺, the β3 17′ His side chain density is absent because these polar residues orient randomly and the pore diameter at this location expands (>4.1 Å) to permit the passage of chloride ions (Pauling radius of 1.8 Å) (Extended Data Fig. 5f–h).

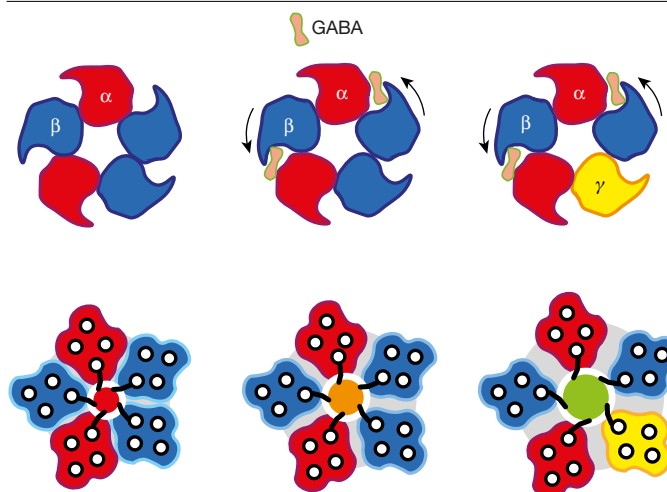

**Fig. 4 | Mode of activation by GABA.** Top-down views of ECDs (top row) and cross sections of TMDs (helices shown as black circles) at the level of the 9′ Leu gate (bottom row) for αβ and αβγ receptors. Pore leucines are represented by black 'fronds' projecting from the innermost α-helix, M2. In response to GABA, the two binding β-subunits are the principal responders, with their ECDs twisting similarly anticlockwise (black arrows) for both αβ receptors and αβγ receptors. The downstream reaction of the TMD is limited in the αβ receptor and the 9′ gate remains mostly closed (red and orange circles), whereas for αβγ receptors, the TMD response is greater and the 9′ gate opens (green).

## Receptor response to GABA

Comparison of the α-CBTx–Zn$^{2+}$, GABA–Zn$^{2+}$ and GABA-bound structures reveals the activation pathway. GABA binding at the two β–α sites induces realignment of the corresponding β-subunit ECDs (chains B and E) and clockwise translation of the β1–β2 and β6–β7 base loops above the transmembrane domain (TMD) (Extended Data Fig. 6a). The motion is equivalent for GABA and GABA–Zn$^{2+}$ structures (Extended Data Fig. 6b, c). Thus, occupation of the pore by Zn$^{2+}$ does not hinder the ECD transition in response to GABA, as previously observed for picrotoxin bound in the channel of αβγ receptors[26,37]. Globally, the GABA-induced motions mirror those observed for αβγ receptors, with the β-subunit ECD in αβ receptors that occupies the 'γ-position' mimicking that of the γ-subunit ECD in αβγ receptors (Extended Data Fig. 6a, d).

At the level of the TMD, without Zn$^{2+}$ bound, the M2–M3 loops (which link the top of channel-lining α-helix 2 to helix 3) of the two GABA-bound β-subunits switch from 'inward' to 'outward' (Fig. 3a; Extended Data Fig. 7a), as previously observed for αβγ receptors[26] (Extended Data Fig. 7b). For the β-subunit occupying the γ-position (chain C), the M2–M3 loop also moves outwards (Fig. 3a), owing to the absence of any inward pull by Zn$^{2+}$, which matches the γ2 subunit in αβγ receptors in both inhibited and GABA-bound conformations (Extended Data Fig. 7b).

With all the M2–M3 loops in the outward position, each of the five channel-lining M2 helices, one contributed by each subunit around the pore, tilt and laterally translate outwards (Extended Data Fig. 8a). The tilt angle increases on average by 0.8° per helix relative to the pore axis, from 3.9° to 4.7° (Extended Data Fig. 8a, b). As a result, the 9′ leucine ring, situated midway along the pore and forming a hydrophobic gate in the α-CBTx–Zn$^{2+}$ inhibited state, retracts to increase the pore diameter from 2 Å to 3.1 Å, and the 9′ C$_\alpha$ perimeter increases from 40.2 Å to 44.2 Å (Fig. 3b–d, Extended Data Fig. 8c). Despite this shift, the pore remains too narrow to permit the passage of chloride anions (Pauling radius of 1.8 Å) and this gate remains closed. This contrasts with GABA-bound αβγ receptor structures, which exhibit open 9′ activation gates with diameters[26,37] in the range 5–6 Å (Fig. 3d, Extended Data Fig. 8c). By comparison with αβ receptors, the M2 outward tilting

for αβγ receptors is consistently greater, increasing from 4.6° to 7.5° for α1β3γ2 and 4.8° to 6.5° for α1β2γ2 in response to GABA binding (Extended Data Fig. 8b). This results in larger 9′ C$_\alpha$ perimeters (45.8 Å for α1β3γ2 and 46.4 Å for α1β2γ2) and hydrophobic Leu side chains are rotated away from the hydrated pore (Extended Data Fig. 8c, d). Indeed, the GABA-bound αβ receptor mean M2 tilt angle (4.7°) does not exceed that of bicuculline-inhibited αβγ receptors (4.7–4.8°; Extended Data Fig. 8b), and the pore profiles are similar (Fig. 3d). The GABA-bound αβ receptor β3 subunit in the γ-position has the biggest difference in tilt angle versus the γ2 subunit (4.4° for α1β3 versus 8.5° for α1β3γ2 and 9.3° for α1β2γ2; Extended Data Fig. 8b, e). Thus, replacing γ2 with the more upright β3 M2 helix has the consequent effect of limiting the outward tilt and expansion of the other subunits, thereby limiting 9′ gate expansion.

Given that extrasynaptic αβ and αβδ receptors have low gating efficacy[7,8,43] (low $P_o$), we hypothesized that this could explain why there is no open 9′ activation gate in our GABA-bound αβ receptor structure. We compared α1β3 versus α1β3γ2 single-channel recordings in the presence of near-saturating (EC$_{95}$) GABA concentrations to evaluate $P_o$. Short and long open dwell times ($\tau_1$ and $\tau_2$) of similar durations were observed for both receptors (α1β3: $\tau_1 = 0.65 \pm 0.15$ ms ($n = 6$) and $\tau_2 = 4.3 \pm 1.1$ ms ($n = 4$; absent from two cells); α1β3γ2: $\tau_1 = 0.78 \pm 0.07$ ms and $\tau_2 = 4.8 \pm 0.7$ ms ($n = 6$) (Extended Data Fig. 9a, b). However, for α1β3 receptors, only 17 ± 6% of openings were of long duration versus 61 ± 3% for α1β3γ2 receptors, confirming a reduced propensity of stable opening for α1β3 channels (Extended Data Fig. 9c). An absence of burst activity for α1β3 receptors precluded measurement of the burst $P_o$ (Methods); nevertheless for patches containing only one apparent ion channel, $P_o$ over the entire course of the recording was significantly lower for α1β3 versus α1β3γ2 receptors (Extended Data Fig. 9d).

Further evidence for reduced channel opening was obtained by measuring the probability of activation[21] ($P_A$) from whole-cell recordings of the maximum response to a saturating concentration of GABA versus GABA plus pentobarbitone, a positive allosteric modulator. If $P_o$ is high in saturating GABA—that is, close to 1—it will not increase further with pentobarbitone. In support of the single-channel data, the $P_A$ for α1β3 (wild type) and for the α1β3 cryo-EM construct were approximately 0.6, compared with 0.94 ± 0.03 for α1β3γ2 ($P < 0.001$) (Extended Data Fig. 10). Our GABA-bound structure is thus consistent with a state in which GABA stabilizes primed ECDs and M2–M3 loops[44,45] to facilitate brief opening, but cannot sufficiently stabilize an open 9′ gate (Fig. 4).

## Conclusion

The structures we present here explain key aspects of the molecular pharmacology of α1β3 GABA receptors, including the mode of α-CBTx antagonism and the signature property for αβ receptors of high-sensitivity Zn$^{2+}$ channel blockade. Despite the ECDs and M2–M3 loops responding to GABA, a more upright β-subunit M2 helix occupying the γ-subunit position results in the 9′ Leu pore gate remaining mostly closed. This provides a molecular explanation for the comparatively low $P_o$ of α1β3 receptors compared with synaptic α1β3γ2 receptors, a feature required to prevent excessive inhibition of neuronal circuitry by αβ and αβδ extrasynaptic subtypes. Given recent successes targeting extrasynaptic GABA$_A$ receptors for therapeutic effects[6], we anticipate that these structures will facilitate future design of drugs that modulate GABA-mediated tonic inhibition.

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

## Methods

### Data reporting

No statistical methods were used to predetermine sample size. The experiments were not randomized and the investigators were not blinded to allocation during experiments and outcome assessment.

### Constructs

The protein sequences used were: human GABA$_A$R α1 (mature polypeptide numbering 1–416, QPSL...TPHQ; Uniprot P14867) and human GABA$_A$R β3 (mature polypeptide numbering 1–447, QSVN...YYVN; Uniprot P28472). The α1 intracellular M3–M4 loop amino acids 313–391 (RGYA...NSVS) were substituted by the SQPARAA sequence[46]. The β3 intracellular M3–M4 loop amino acids 308–423 (GRGP...TDVN) were substituted by a modified SQPARAA sequence containing the *Escherichia coli* soluble cytochrome B562RIL41 (BRIL, amino acids 23–130, ADLE...QKYL, Uniprot P0ABE7) to give an M3–M4 loop with the sequence SQPAGTBRILTGRAA, necessary to boost protein yields[46]. The mature engineered α1 construct with a 1D4 purification tag derived from bovine rhodopsin (TETSQVAPA) that is recognized by the Rho-1D4 monoclonal antibody (University of British Columbia)[47,48] was cloned into the pHLsec vector after the vector secretion signal[49], ending TPHQGTTETSQVAPA. An alternative tagged version of the engineered α1 construct was cloned into the pHLsec vector after the secretion signal with an N-terminal streptavidin binding protein (SBP) and TEV cleavage site, starting (GCVA) with EMDEKTTGWRGGHVVEGLAGELEQLR ARLEHHPQGQREPDYDIPTTENLYFQGTG-GABRα1(QPSL...), and ending with a stop codon and no C-terminal tag. The engineered β3 construct was cloned into pHLsec after the vector secretion sequence without any tags.

### Expression and protein preparation

Four-hundred millilitres of HEK 293S-GnTI- cells (which yield proteins with truncated N-linked glycans, Man$_5$GlcNAc$_2$[50,51] were grown in suspension up to densities of $2 \times 10^6$ cells per ml in Protein Expression Media (PEM) (Invitrogen) supplemented with L-glutamine, non-essential amino acids (Gibco) and 1% v/v fetal calf serum (Sigma-Aldrich). Cultures were grown in upright round 1-l bottles with filter lids, shaking at 130 rpm, 37 °C, 8% CO$_2$. For transient transfection, cells were collected by centrifugation (200$g$ for 5 min) and resuspended in 50 ml Freestyle medium (Invitrogen) containing 0.6 mg PEI Max (Polysciences) and 0.2 mg plasmid DNA, followed by a 4 h shaker-incubation in a 2-l conical flask at 160 rpm. Plasmids were transfected at 1:1:4 ratio (that is, 0.035:0.035:0.13 mg) of α1-1D4:SBP-α1:β3. Subsequently, culture medium was topped up to 400 ml with PEM containing 1 mM valproic acid and the cell suspension was returned to empty bottles. Typically, 40–50% transfection efficiency was achieved, as assessed by inclusion of 3% DNA of a control GFP plasmid. Seventy-two hours after transfection, cell pellets were collected, snap-frozen in liquid N$_2$ and stored at −80 °C.

The receptor was double purified against first the SBP tag and then the 1D4-tag to only purify receptors containing one of each of the alternatively SBP or 1D4 tagged α1 subunits. The β3 subunit was transfected in excess relative to the α1 subunit, at 2:1, to ensure that the double-purified material consisted of only receptors comprising two α1 subunits and three β3 subunits, as previously proposed[52,53]. Note that 1 mM histamine was included in all the buffers described below, throughout the purification to aid yield. The cell pellet (approx. 7–10 g) was solubilized in 30 ml buffer containing 20 mM HEPES pH 7.2, 300 mM NaCl, 1% (v/v) mammalian protease inhibitor cocktail (Sigma-Aldrich, cat.P8340) and 1.5% (w/v) lauryl maltose neopentyl glycol (LMNG, Anatrace) at a 10:1 molar ratio with cholesterol hemisuccinate (CHS, Anatrace), for 2 h at 4 °C. Insoluble material was removed by centrifugation (10,000$g$, 15 min). The supernatant was diluted twofold in a buffer containing 20 mM HEPES pH 7.2, 300 mM NaCl and incubated for 2 h at 4 °C with 1 ml high-capacity streptavidin beads (Thermofisher 20361). Affinity-bound samples were washed by gravity flow for 30 min at 4 °C with 10 ml of detergent-lipid (DL) buffer containing 20 mM HEPES pH 7.2, 300 mM NaCl, and 0.1% (w/v) LMNG 10:1 CHS containing an excess (400 μl) of a phosphatidylcholine (POPC, Avanti) and bovine brain lipid (BBL) extract (type I, Folch fraction I, Sigma-Aldrich) mixture (POPC:BBL ratio 85:15). POPC and BBL extract stocks (10 and 20 mg ml$^{-1}$, respectively) were prepared by solubilization in 3% w/v dodecyl maltopyranoside (DDM). Protein was eluted in 2 ml DL buffer supplemented with 5 mM biotin, for 2 h at 4 °C. The elution was incubated for 2 h at 4 °C with 100 μl CNBr-activated sepharose beads (GE Healthcare) pre-coated with Rho-1D4 antibody (British Columbia) (3.3 g dry powdered beads expand to approximately 10 ml during coupling of 50 mg of 1D4 antibody in 20 ml phosphate buffered saline). The beads were gently centrifuged (300$g$, 5 min) and washed with 10 ml of DL buffer.

On-bead nanodisc reconstitution was performed[26], in which the beads were equilibrated with 1 ml of DL buffer. Beads were centrifuged and excess solution removed leaving 100 μl DL buffer, which was topped up with 75 μl of MSP2N2 at 5 mg ml$^{-1}$ together with Bio-Beads (40 mg ml$^{-1}$ final concentration) and incubated for 2 h rotating gently at 4 °C. The MSP2N2 belt protein was produced as previously described[24]. After nanodisc reconstitution, the 1D4 resin and Bio-Bead mixture was washed extensively with buffer (300 mM NaCl, 50 mM HEPES pH 7.6) to remove empty nanodiscs. Protein was eluted using 100 μl of buffer containing 75 mM NaCl, 12.5 mM HEPES pH 7.6, 500 μM 1D4 peptide overnight with gentle rotation at 4 °C. The next day, beads were centrifuged and the eluate was collected, which contained protein at 0.3 mg ml$^{-1}$. This was used directly for cryo-EM grid preparation. Purified Mb25[28] was added at a twofold molar excess. For drug treatments, GABA was added at 200 μM, ZnCl$_2$ at 20 μM and α-CBTx (Smartox) at 10 μM. For the α-CBTx–Zn$^{2+}$ (3.0 Å resolution) and GABA–Zn$^{2+}$ (2.8 Å resolution) structures a concentration of 20 μM Zn$^{2+}$ was chosen because it is sufficient to achieve approximately 90% inhibition (Extended Data Fig. 5a) while minimizing risks of off-target binding to low-affinity Zn$^{2+}$ sites[9]. For grid preparation, 3.5 μl of sample was applied onto glow-discharged gold R1.2/1.3 300 mesh UltraAuFoil grids (Quantifoil) for and then blotted for 5.5 s at blot force of −15 before plunge-freezing the grids into liquid ethane cooled by liquid nitrogen. Plunge-freezing was performed using a Vitrobot Mark IV (Thermo Fisher Scientific) at approximately 100% humidity and 14.5 °C.

### Nb25 purification and production

Nb25 was produced exactly as described[29], and reproduced here. Nb25 was produced and purified in milligram quantities from WK6su *E. coli* bacteria. Bacteria were transformed with about 200 ng of the nanobody expression plasmid pMESy4 containing Nb25 and selected on lysogeny broth (LB)-agar plates containing 2% glucose and 100 μg ml$^{-1}$ ampicillin. Two or three colonies were used to prepare a preculture, which was used to inoculate 0.5 l Terrific broth (TB) cultures supplemented with 0.1% glucose, 2 mM MgCl2 and 100 μg ml$^{-1}$ ampicillin. Cultures were grown at 37 °C until their absorbance at 600 nm reached 0.7, at which point Nb25 expression was induced with 1 mM IPTG. After induction, cells were grown at 28 °C overnight and harvested by centrifugation (20 min, 5,000$g$). Nanobodies were released from the bacterial periplasm by incubating cell pellets with an osmotic shock buffer containing 0.2 M Tris, pH 8.0, 0.5 mM EDTA and 0.5 M sucrose. The C-terminally His$_6$-tagged Nb25 was purified using nickel-affinity chromatography (binding buffer: 50 mM HEPES, pH 7.2, 1 M NaCl, 10 mM imidazole; elution buffer: 50 mM HEPES, pH 7.2, 0.2 M NaCl, 0.5 M imidazole) and then subjected to size-exclusion chromatography on a Superdex 75 16/600 column (GE Healthcare) in 10 mM HEPES, pH 7.2, 150 mM NaCl. Nb25 stocks were concentrated to 5–10 mg ml$^{-1}$, snap frozen in liquid nitrogen and stored at −80 °C. Yield was in the range 2–10 mg from 500 ml bacterial suspension.

## Cryo-electron microscopy data acquisition and image processing

All cryo-EM data presented here were collected in the Department of Biochemistry, University of Cambridge and all data collection parameters are given in Extended Data Table 1. Krios data were collected using FEI EPU and then processed using Warp[54] and cryoSPARC[55,56]. In short, contrast transfer function correction, motion correction and particle picking were performed using Warp. These particles were subjected to 2D classification in cryoSPARC followed by ab initio reconstruction to generate the initial 3D models. Particles corresponding to different classes were selected and optimized through iterative rounds of heterogeneous refinement as implemented in cryoSPARC. The best models were then further refined using homogeneous refinement and finally non-uniform refinement in cryoSPARC. For the final reconstructions the overall resolutions were calculated by FSC at 0.143 cutoff (Extended Data Table 1). A local_res map was generated in cryoSPARC using the program 'local resolution estimation'. The resolution range was based on the Fourier shell correlation output calculated for voxels only within the mask output from the homogenous refinement job used as the input for local resolution estimation. To generate maps coloured by local resolution, the local_res map along with the main map were opened in UCSF Chimera[57] and processed using the surface colour tool.

## Model building, refinement, validation, analysis and presentation

Model building was carried out in Coot[58] using PDB 6HUO as a template for the GABA$_A$R α1β3 GABA map. The model was docked into the cryo-EM density map using the dock_in_map program, PHENIX suite[59]. The map resolution was sufficient to allow ab initio building of M3–M4 helix linkers for GABA$_A$R α1. Before refinement, phenix_ready_set was run to generate the restraints for the bound ligands including lipids, GABA and histamine and optimize the metal ion coordination restraints. The geometry constraint files for small-molecule ligands used in the refinement were generated using the Grade Web Server (Global Phasing). The model was improved iteratively by rounds of refinement using phenix_real_space_refine and manual inspection and improvement of refined models in Coot. Model geometry was evaluated using the MolProbity Web Server[60]. The new GABA$_A$R α1β3 GABA model was subsequently used as a template in the GABA–Zn$^{2+}$ and αCBTx–Zn$^{2+}$ maps, which were then modified and built using the same process as applied for creating the GABA map. PDB:1YI5 Chain J was used as a template for α-CBTx. Phenix_mtriarge[61] was used to calculate the resolution at 0.5 FSC. Pore permeation pathways and measurements of pore diameters were generated using the HOLE plug-in[62] in Coot. Structural overlays were generated using Matchmaker function in UCSF chimera[57] and C$_α$ r.m.s.d. values measured using the rmsd function. Rotation angles were calculated using UCSF Chimera. Structural presentations for figures were produced using UCSF Chimera or Pymol (Schrödinger).

## Electrophysiology

**Whole-cell recordings.** Whole-cell responses were recorded in patch clamp experiments from HEK 293 cells transiently transfected with human GABA$_A$ α1β3 (WT), α1$_{GLIVI}$β3$_{BRIL}$ (α1β3cryo-EM; see 'Constructs') or α1β3γ2 (WT). HEK 293 cells were grown in DMEM supplemented with 10% v/v fetal bovine serum, 100 U ml$^{-1}$ penicillin-G and 100 μg ml$^{-1}$ streptomycin (37 °C; 95% air/5% CO$_2$), and transfected using a calcium-phosphate precipitation method with α1:β3:GFP or α1:β3:γ2:GFP cDNAs in a ratio of 1:1:1 or 1:1:3:1, respectively, 12–24 h before experimentation. Recordings were performed with cells continuously perfused with Krebs solution composed of (mM): 140 NaCl, 4.7 KCl, 1.2 MgCl$_2$, 2.52 CaCl$_2$, 11 Glucose and 5 HEPES (pH 7.4; ~300 mOsm). Patch pipettes (TW150F-4; WPI; 3–4 MΩ) were filled with an internal solution containing (mM): 140 KCl, 1 MgCl$_2$, 11 EGTA, 10 HEPES, 1 CaCl$_2$, 2 K-ATP (pH 7.2; ~305 mOsm). Drugs were

applied to cells using fast Y-tube application, where Zn$^{2+}$ and histamine were pre-applied before co-application with GABA. Cells were voltage-clamped at −40 mV with an Axopatch 200B amplifier (Molecular Devices), currents were digitized at 50 kHz via a Digidata 1322A (Molecular Devices), filtered at 5 kHz (−36 dB), and acquired using Clampex 10.2 (Molecular Devices). Series resistance was compensated at 60–70% (lag time 10 μs).

For free Zn$^{2+}$ concentration experiments, the Zn$^{2+}$ chelator tricine was used to precisely control Zn$^{2+}$ concentration and eliminate background Zn$^{2+}$ contamination. Krebs solution was supplemented with 10 mM tricine and pH corrected to 7.4. The free Zn$^{2+}$ concentrations were calculated according to the equation: [Zn]$_{free}$ = ($α × K_d ×$ [Zn]$_{total}$)/[tricine]; where [Zn]$_{total}$ is known, $K_d$ is dissociation constant, $α$ is 6.623777 (that is, 1 + ([H$^+$; $M = 3.98 × 10^{-8}$]/[association constant ($K_a$) for tricine; $M = 7.08 × 10^{-9}$]) ($M$, molar), $K_d$ for tricine is 10 μM, and the concentration of tricine was 10 mM. In these experiments we used the following total Zn$^{2+}$ concentrations (μM): 1, 3, 10, 30, 100, 300, 1,000, 3,000 and 10,000, which in 10 mM tricine-buffered Krebs solution, resulted in the following calculated free-zinc concentrations (nM): 6.6, 19.8, 66, 198, 662, 1,990, 6,600, 19,900 and 66,200.

**Data analysis for whole-cell recordings.** Peak current responses and desensitization rates were obtained using Clampfit 10.2 (Molecular Devices). The EC$_{50}$ and IC$_{50}$ values were obtained by curve fitting concentration response data from individual experiments to the Hill equation ($I/I_{max} = A^n/(EC_{50}{}^n + A^n)$) or inhibition equation ($I/I_{max} = 1 − (B^n/(IC_{50}{}^n + B^n))$, where $A$ is GABA or histamine concentration, $B$ is Zn$^{2+}$ concentration and $n$ is the Hill coefficient; data were fitted using Origin 6.0. Potency values are presented as pEC$_{50}$ or pIC$_{50}$ with s.e.m., and the mean was converted into a molar concentration (pEC$_{50}$ = −log EC$_{50}$; pIC$_{50}$ = −log IC$_{50}$). Experiments were repeated at least three times from three different cells. Statistical analysis and graphical data presentations were performed using Prism 9 (GraphPad Software). Unpaired two-tailed Student's $t$-tests were used for single comparisons of properties between wild-type and the Cryo-EM construct, and no values reached significance; that is, none were less than 0.05 (values reported in relevant figure legends). For comparing the two Zn$^{2+}$ inhibition concentrations across wild-type and cryo-EM constructs a one-way ANOVA and Tukey multiple comparisons post hoc test was used, and showed no significant differences across groups, $F_{(3, 24)} = 0.6449$; $P = 0.5937$. Specific statistical analyses performed for each dataset comparison are provided in the relevant figure legends.

**Single-channel recording.** Single GABA-activated channel currents were recorded in outside-out patches from transfected HEK 293 cells at −70 mV holding potential. Channel currents were recorded using an Axopatch 200B and filtered at 5 kHz (4-pole Bessel filter) before digitizing at 20 kHz with a Digidata 1322A. The fixed time resolution of the system was set at 80 μs. WinEDR was used for analysing single channel data. The single-channel current was determined from compiling channel current amplitude histograms and fitting Gaussian components to define the mean current, s.d. and the total area of the component. The single-channel conductance was calculated from the mean unitary current and the difference between the patch potential and GABA current reversal potential. Individual open and closed dwell times were measured using a 50% threshold cursor applied to the main single channel current amplitude in each patch. The subsequent detection of open and closed events formed the basis of an idealized single channel record used for compiling the dwell time distributions. Frequency distributions were constructed from the measured individual open and closed times and analysed by fitting a mixture of exponentials, defined by:

$$y(t) = \sum_{i=1}^{n} \frac{A_i}{\tau_i} × e^{(-t/\tau_i)}$$

where $A_i$ is the area of the $i$th component to the distribution and $\tau_i$ represents the corresponding exponential time constant. A Levenberg–Marquardt non-linear least-squares routine was used to determine the values of individual exponential components. An $F$-test determined the optimal number of exponential components that were required to fit the individual dwell time distributions. The determination of a critical closed time ($\tau_{crit}$) to define bursts of GABA channel activity was performed as previously described[63]. Given that sufficient numbers of bursts were not resolved for $\alpha1\beta3_{WT}$, we could not compare intra-burst open probabilities. Therefore, we assessed the open probabilities from continuous single channel recordings where there was no evidence of channel stacking during GABA application and thus it was possible, but not guaranteed, that these patches contained only one active channel. Even if this premise is false, the same analysis conditions were applied to recordings for both $\alpha1\beta3\gamma2_{WT}$ and $\alpha1\beta3_{WT}$ receptors. To ensure near-accurate estimates of GABA channel open probability patches were rejected if they displayed multiple channel activation or if such activity accounted for more than 2% of the open-channel currents measured in a single recording. The single channel current for $\alpha1\beta3\gamma2_{WT}$ of about 1.9 pA at −70 mV, reflected a main state conductance equivalent to 28 pS, whereas for $\alpha1\beta3_{WT}$ the main open state current and conductance were lower as expected[8], at about 1.3 pA at −70 mV, equivalent to about 19 pS.

## Cell lines

HEK 293T cells used for electrophysiology and HEK 293S GnTI⁻ cells used for protein production for cryo EM were obtained from ATCC. Further authentication of cell lines was not performed for this study. Mycoplasma testing was not performed for this study.

## Reporting summary

Further information on research design is available in the Nature Research Reporting Summary linked to this paper.

## Data availability

Atomic model coordinates for $\alpha$-CBTx–$Zn^{2+}$, GABA–$Zn^{2+}$ and GABA-bound structures have been deposited in the Protein Data Bank with accession codes 7PC0, 7PBZ and 7PBD, respectively. Cryo-EM density maps have been deposited in the Electron Microscopy Data Bank with accession codes EMD-13315, EMD-13314 and EMD-13290 respectively.

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

**Acknowledgements** We thank L. Cooper for training in cryo-EM grid preparation and for performing grid clipping; J. Stayaert and R. Aricescu for advice on, access to and use of Mb25; R. Aricescu and S. Masiulis for valuable discussions and training in sample preparation; and E. Seiradake for provision of the pHLsec-SBP plasmid. This work was supported by a Department of Pharmacology new lab start-up fund, the University of Cambridge Isaac Newton and Wellcome Trust Institutional Strategic Support Fund, and Academy of Medical Sciences Springboard Award (SBF004\1074). Electrophysiology work in the T.G.S. laboratory was funded by an MRC programme grant (MR/T002581/1) and Wellcome Trust Collaborative Award (217199/Z/19/Z). The cryo-EM facility receives funding from the Wellcome Trust (206171/Z/17/Z; 202905/Z/16/Z) and University of Cambridge.

**Author contributions** V.B.K. performed protein purification, atomic model building and structural interpretation. A.A.W. performed protein purification. M.M. and T.G.S. designed and analysed the electrophysiological experiments, which were performed by M.M. and V.D. S.W.H. and D.Y.C. performed cryo-EM data acquisition and processing. P.S.M. performed construct design, protein purification, cryo-EM sample preparation, atomic model building and structural interpretation. P.S.M. wrote the manuscript with input from all other authors.

**Competing interests** The authors declare no competing interests.

**Additional information**
**Correspondence and requests for materials** should be addressed to Trevor G. Smart or Paul S. Miller.

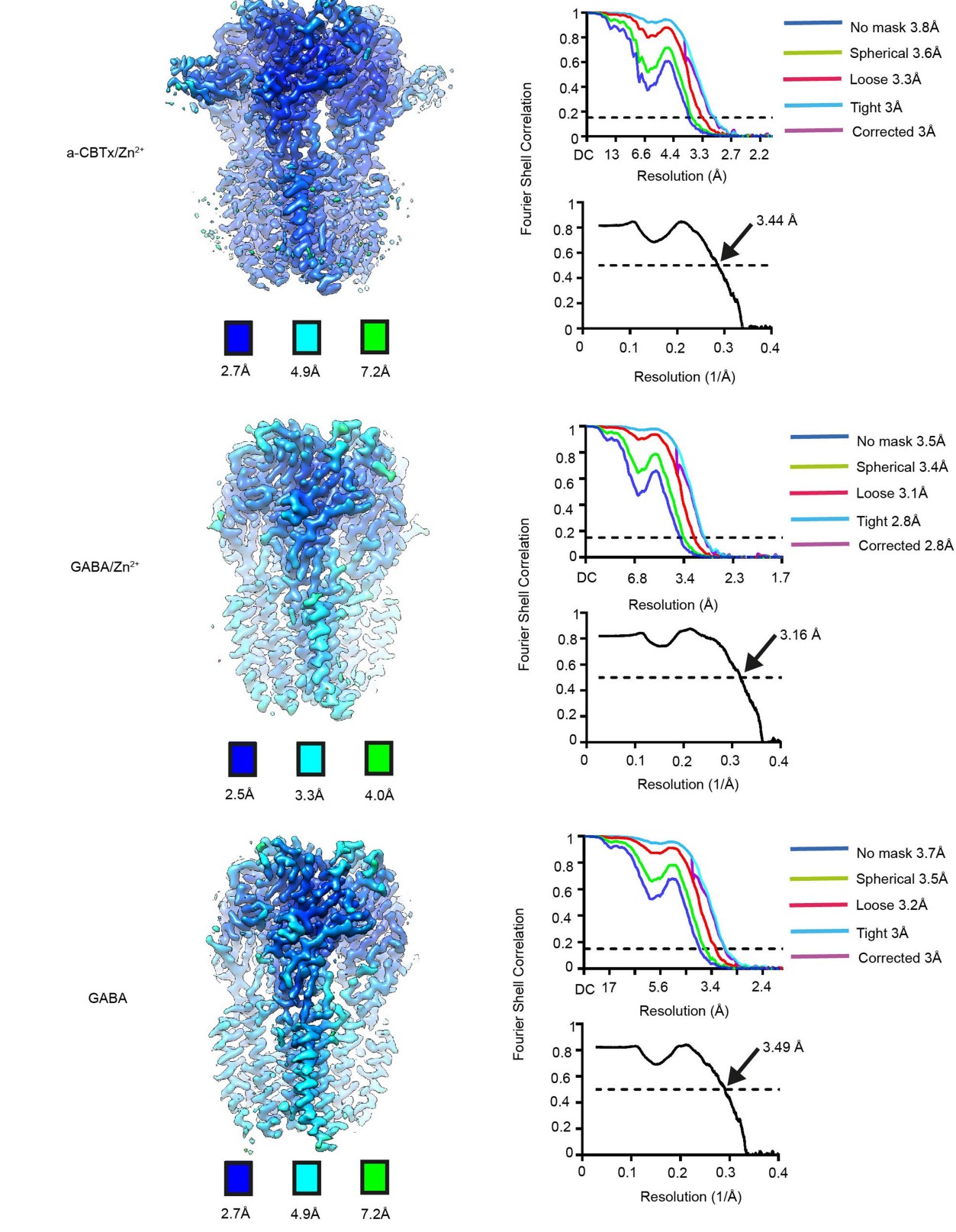

**Extended Data Fig. 1 | Local resolution maps, overall plotted resolutions, and global map-model agreements.** For the three structures, α-CBTx/Zn²⁺, GABA/Zn²⁺, and GABA-bound, a map on the left is coloured by local resolution (see methods). Maps of Fourier shell correlation (FSC) (upper right panels) and map-model FSC (lower right panels) plots are also shown. Relevant statistics for these maps are presented in Extended Data Table 1.

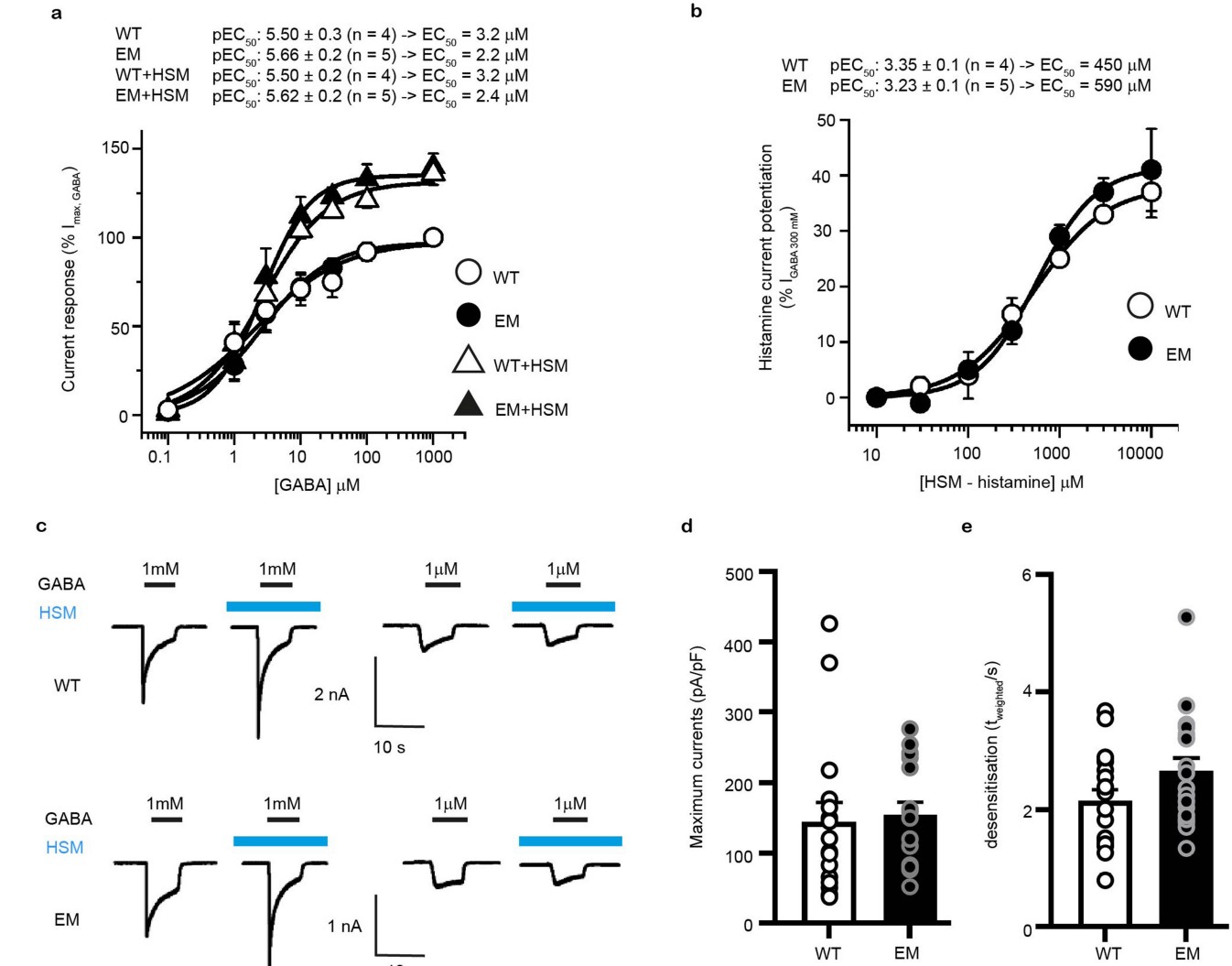

**Extended Data Fig. 2 | GABA responses and histamine potentiation. a**, GABA concentration response curves for $\alpha\beta_{WT}$ (white symbols) and $\alpha\beta_{CryoEM}$ (black symbols) in the absence (circles) or presence (triangles) of 3 mM histamine (HSM). Data was obtained in whole cell patch clamp experiments performed on transiently transfected HEK 293 cells. **b**, Histamine concentration response curves for potentiating the 300 µM GABA response for $\alpha\beta_{WT}$ and $\alpha\beta_{CryoEM}$. For **a**, and **b**, points represent mean ± s.e.m. Curves generated are $n = 4$ and $n = 5$ for WT and EM constructs respectively. For **a**, One-way ANOVA showed no statistical difference across the 4 $pEC_{50}$ values, and for **b**, Two-sided unpaired t-test showed values were not statistically different, P = 0.51.

**c**, Representative whole-cell patch clamp current responses to GABA and GABA + 3 mM histamine applications. Note that histamine was pre-applied before co-applying with GABA (blue lines). **d–e**, Bar charts showing average maximum (1 mM) GABA current response levels and rates of desensitisation, respectively, for $\alpha\beta_{WT}$ and $\alpha\beta_{CryoEM}$. Individual values are shown as circles, and bars are means ± s.e.m. EM $I_{max}$ $n = 14$, WT $I_{max}$ $n = 15$, EM desens $n = 11$, WT desens $n = 10$. Two-sided unpaired t-test showed values were not statistically different for either property, P = 0.64 and 0.08, respectively. Each $n = 1$ value of an $pEC_{50}$, $I_{max}$ and desensitisation value were from biologically independent patch-clamp experiments from individual cells.

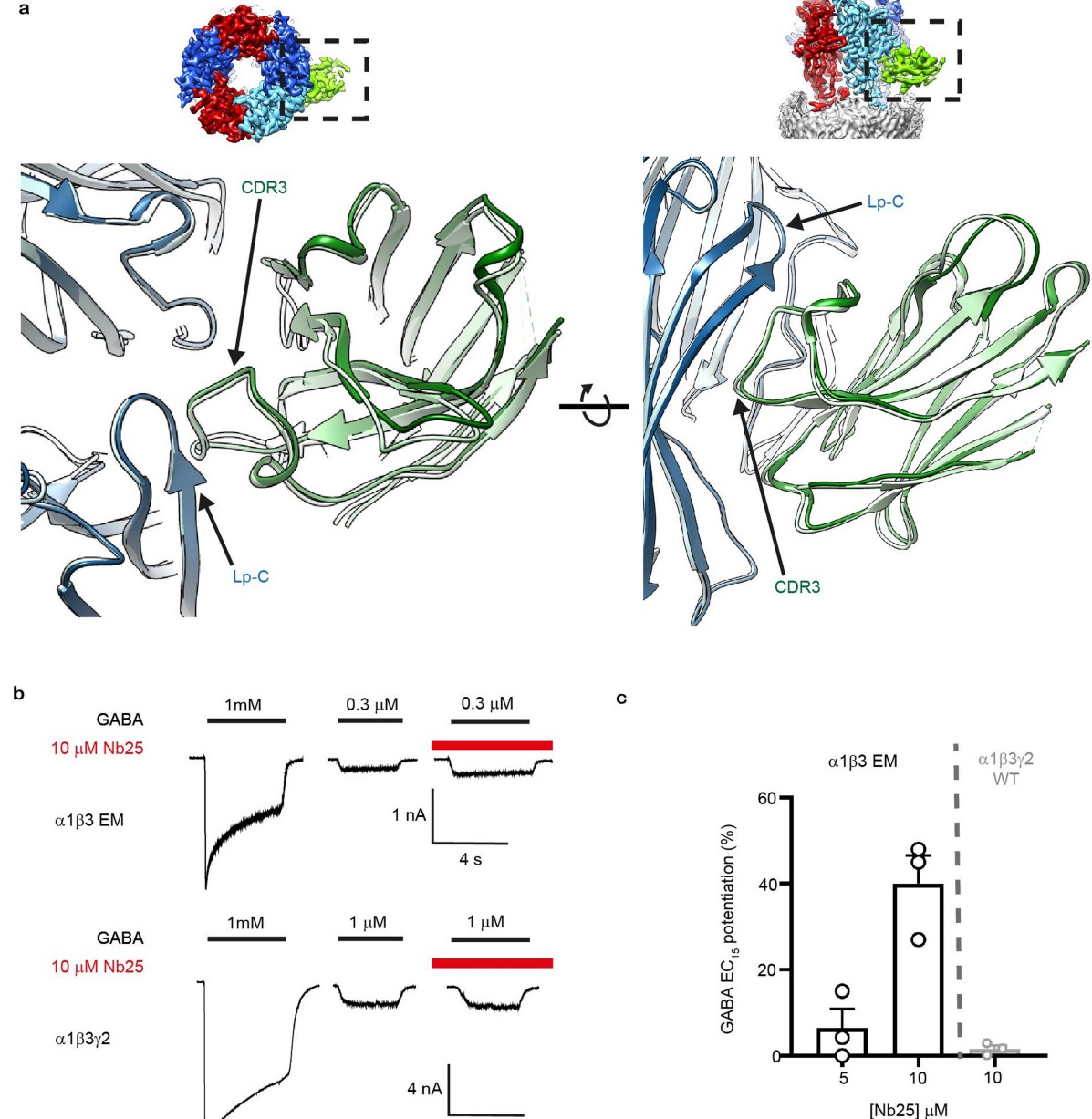

**Extended Data Fig. 3 | Nb25 potentiation at α1β3 receptors. a**, Atomic models show no obvious distinctions for the β-β interface and the Nb25 binding pose of α1β3 receptors in the inhibited α-CBTx/Zn²⁺-bound conformation (3.0 Å, darker shades) versus the GABA-bound conformation (3.0 Å, lighter shades), consistent with any functional impacts exerted by Nb25 being subtle. Upper insets are viewing aids to highlight the region of the protein complex being viewed. Nb in green, β-subunits in blue, α-subunits in red. CDR3 is complementarity determinant loop 3 of Nb25. **b**, Representative currents of whole cell patch clamp responses to GABA and GABA + 10 μM Nb25 applications for α1β3$_{CryoEM}$ versus α1β3γ2 wild-type. Note that Nb25 was pre-applied before co-applying with GABA (red lines). **c**, Bar chart showing average potentiation of EC$_{15}$ GABA current responses for αβ$_{CryoEM}$ versus α1β3γ2 wild-type by Nb25, revealing a weak selective potentiation of α1β3 receptors due to the β-β interface, which is absent from α1β3γ2 receptors. Bars are means ± s.e.m. $n = 3$, each value being from biologically independent patch-clamp experiments from individual cells.

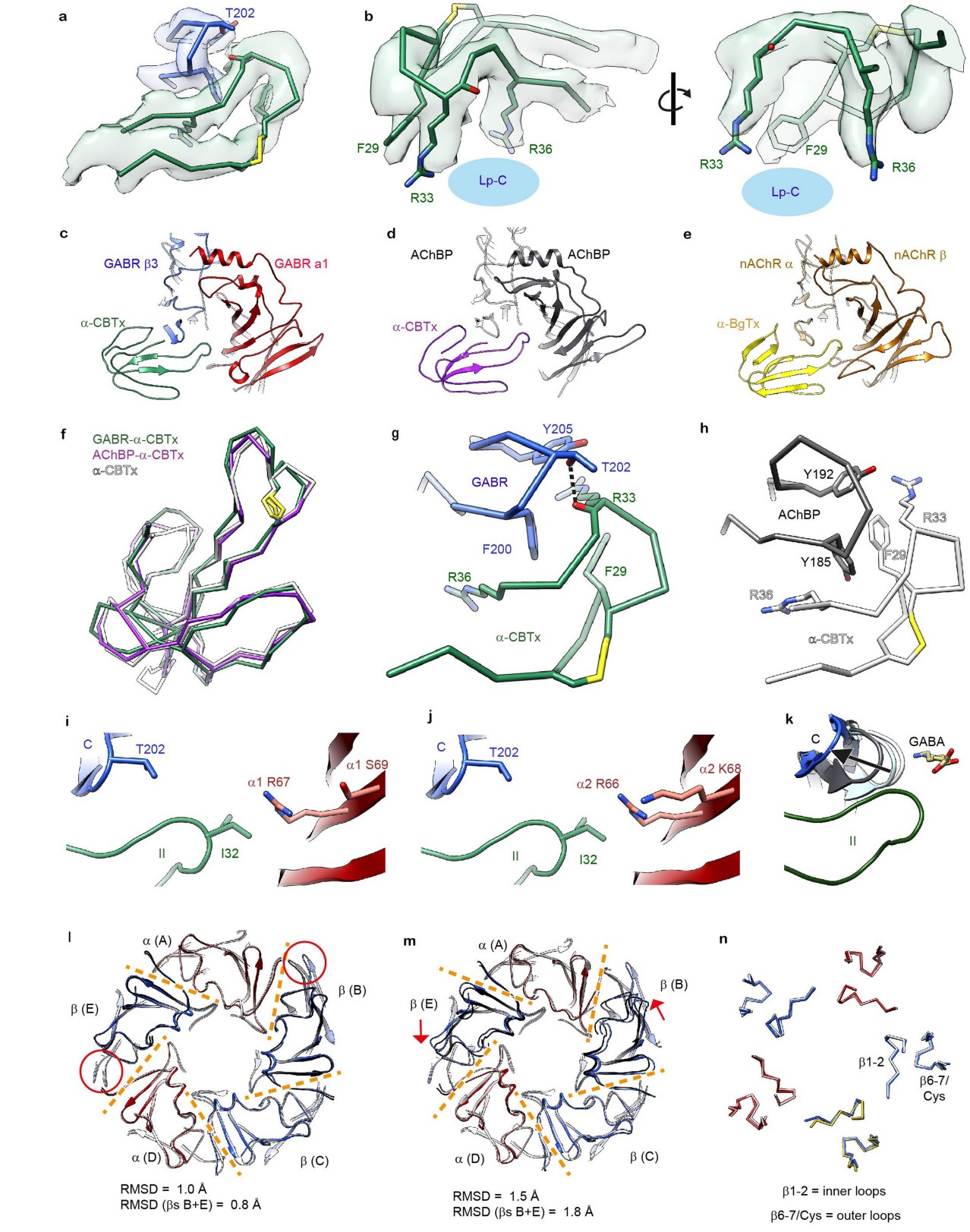

**Extended Data Fig. 4** | See next page for caption.

**Extended Data Fig. 4 | α-Cobratoxin binding mode and αβ receptor conformation. a**, Atomic model fit in the cryo-EM map density of the α1β3 receptor bound by α-CBTx/$Zn^{2+}$ (3.0 Å) for the receptor β3 subunit loop-C (blue) and toxin finger II (green); yellow segment is Cys26-Cys30 side chain Cys-bridge. **b**, Alternative view of the toxin to show side chain density for Phe29, and binding residues Arg33 and Arg36. **c**–**e**, Atomic models showing common binding poses for toxins against pLGICs at inter-subunit interfaces (AChBP PDB 1YI5; nAChR PDB 6UWZ). **f**, Overlays showing closely matching arrangements of α-CBTx atomic models for GABRα1β3, AChBP and apo-α-CBTx (PDB 1ZFM). **g**, **h**, Atomic models comparing α-CBTx finger II binding mode to GABA$_A$ receptor β3 subunit loop-C versus AChBP. **i**, Receptor α1-subunit Arg67 side chain can move away from toxin finger II Ile32 to accommodate toxin binding. **j**, Mechanism of reduced toxin sensitivity for α2-GABA$_A$R subunit, caused by Lys68 in the equivalent position to α1 Ser69, which can be explained by Lys68 sterically and electrostatically hindering Arg67 movement away from toxin Ile32 to reduce accommodation of the toxin. **k**, Overlays to compare loop-C outward motion imposed by α-CBTx on αβ receptor (arrow; GABA-bound in pale blue, toxin-bound blue), and bicuculline on αβγ receptor (GABA-bound in white PDB:6HUO, bicuculline-bound in dark grey PDB:6HUK; bicuculline not shown). **l**, Overlays of cross-section of top of pentameric ECD for α1β3 α-CBTx/$Zn^{2+}$ (blue/red) versus an inhibited state of α1β3γ2 bound by the specific antagonist bicuculline (grey). The only distinguishable difference is the exaggerated outward translation of the β3-subunit loop-C for α-CBTx/$Zn^{2+}$ (red circles) caused by toxin binding (toxin not shown). Inter-subunit interfaces are indicated by orange dashed lines. **m**, Same as **l** except overlay of αβ versus GABA-activated state of α1β3γ2 to reveal a greater divergence in conformation (RMSD increased from 1.0 Å in **l** to 1.5 Å in **m**), in particular for the agonist-responding β-subunits (chains B/E), indicated by red arrows (RMSD increases to 1.8 Å). **n**, Overlays of the β1-β2 loops (inner) and β6-β7 loops (Cys-loops; outer) at the base of the ECD, which oppose the TMD, for α1β3 α-CBTx/$Zn^{2+}$ (blue/red) versus the inhibited state of α1β3γ2 bound by bicuculline (pale shades, yellow for γ2 loops), showing occupation of the same positions for all subunits, including for the α1β3 Chain C β-subunit fitting the α1β3γ2 Chain C γ2-subunit position.

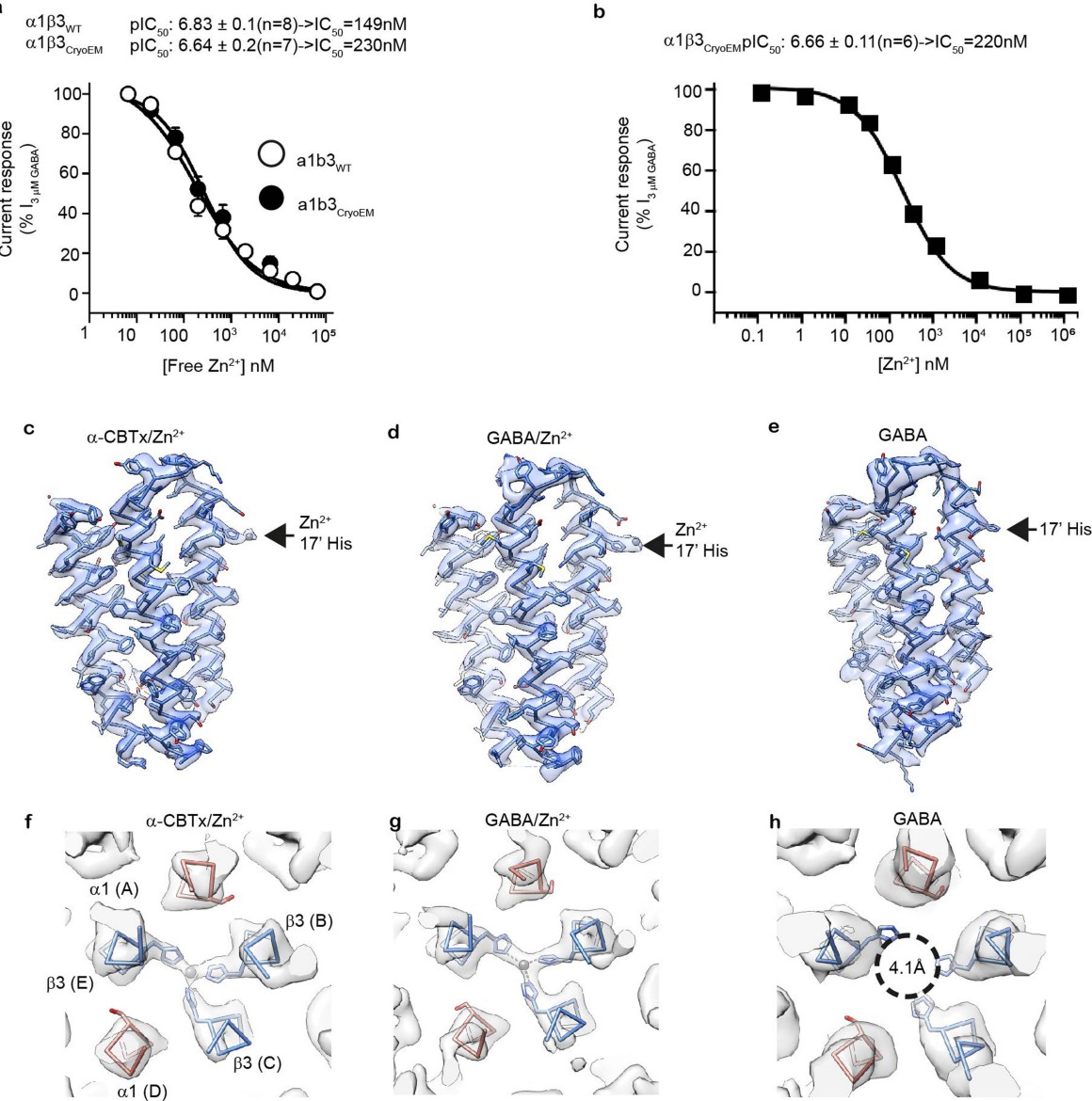

**Extended Data Fig. 5 | Zn²⁺ inhibition. a**, Free* Zn²⁺ inhibition curves from whole cell patch-clamp experiments performed in recording buffer supplemented with 10 mM tricine for αβ$_{WT}$ (white symbols) and αβ$_{CryoEM}$ (black symbols) expressed in HEK293 cells. Points represent mean ± s.e.m. Curves generated are $n = 8$ and $n = 7$ respectively of WT and EM constructs, of biologically independent patch-clamp experiments from individual cells. Two-sided unpaired t-test showed pIC50 values were not statistically different, P = 0.35. **b**, similar Zn²⁺ inhibition curve for αβ$_{CryoEM}$, but in the absence of Zn²⁺-chelating tricine, showing that contaminating Zn²⁺ in buffers was not impacting sensitivity in any way, $n = 6$. **c**–**e**, Atomic model fits in cryo-EM map density of β3 (chain E) subunit TMDs

for α-CBTx/Zn²⁺ (3.0 Å), GABA/Zn²⁺ (2.79 Å) and GABA (3.04 Å) respectively. Zn²⁺ density at 17' His is indicated. Note the 17' density is absent when Zn²⁺ is not bound. **f**–**h**, Atomic model fits in cryo-EM map density for top-down slices of the pentamer at the 17' pore position for α-CBTx/Zn²⁺, GABA/Zn²⁺ and GABA-bound structures, respectively. **h**, Density for 17' His residues is absent when Zn²⁺ is not bound to coordinate them, indicating these side chains are highly mobile. Pore expands nominally to 4.1 Å diameter (variable depending on flexible His arrangement). *free Zn²⁺ concentration controlled and determined using the chelator, tricine (see "Methods").

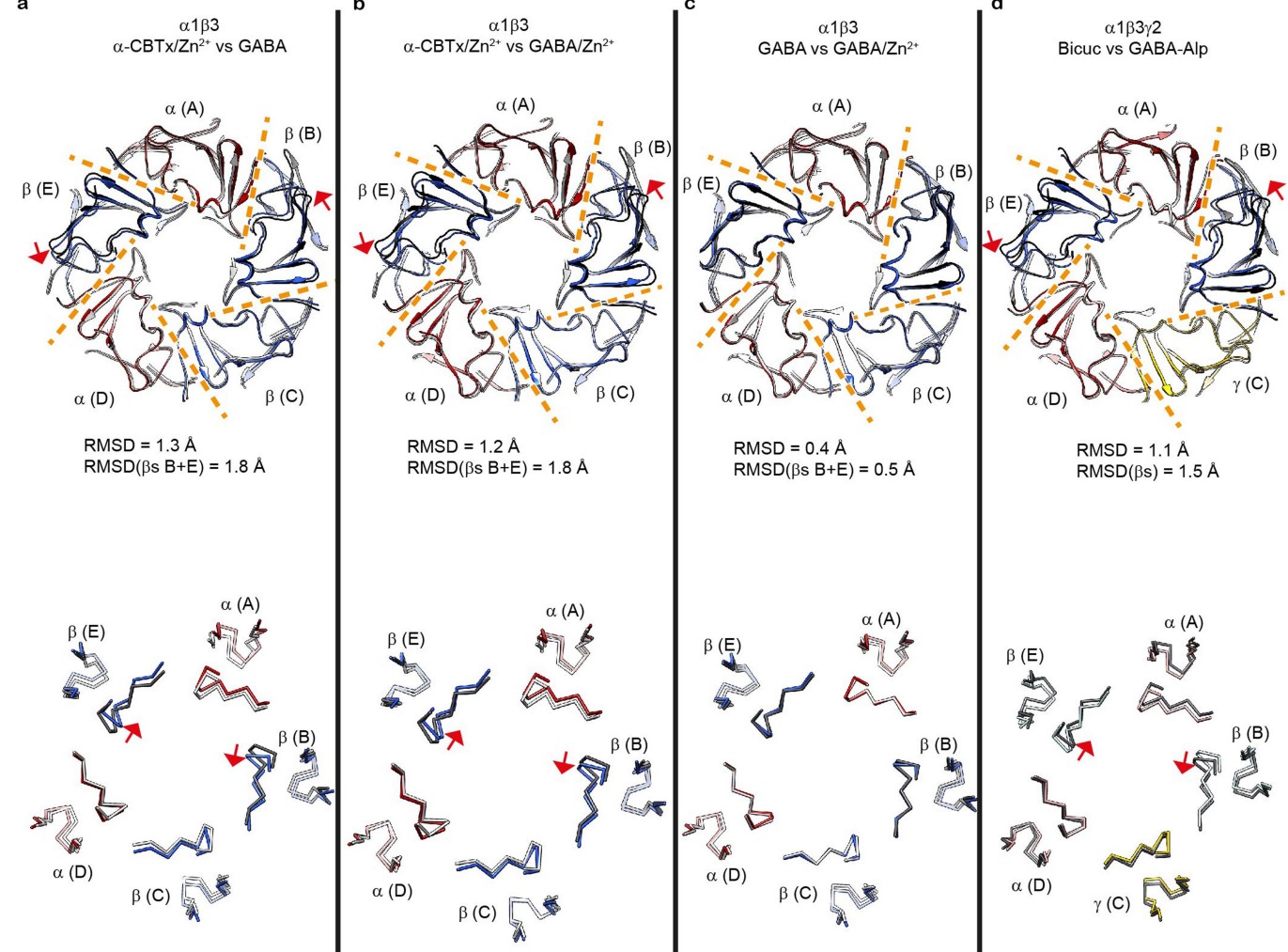

**Extended Data Fig. 6 | Impact of GABA binding on ECD conformation.**
**a**, Upper panel: Overlay cross-sections of the top of pentameric ECDs of
α-CBTx/Zn²⁺ (grey) versus GABA (α-red/β-blue)-bound atomic models.
Greatest divergence is observed for the GABA binding β3-subunits (chains B/E),
which have tilted/rotated in response to GABA binding, red arrows, and is
reflected by RMSD being higher for these subunits, 1.8 Å, relative to the whole
ECD, 1.3 Å. **a**, Lower panel: Overlay of the β1-β2 loops (inner) and β6-β7 loops
(Cys-loops; outer) at the base of the ECD which oppose the TMD (not shown).
This shows the resultant translation for the GABA binding β3-subunits (chains
B/E; red arrows), caused by the motion in the upper ECD (upper panel). **b**, same

as **a**, but for α-CBTx/Zn²⁺ (grey) versus GABA/Zn²⁺ (α-red/β-blue). Differences
are the same because GABA induces the same ECD motions even with Zn²⁺
bound in the pore. **c**, same as **a**, but for GABA (α-red/β-blue) versus GABA/Zn²⁺
(grey). As these ECDs have undertaken the same motions in response to
binding GABA, RMSDs are lower and β-subunit RMSDs do not increase relative
to whole ECD. **d**, same as **a**, but for α1β3γ2 bicuculline-bound (grey; PDB 6HUK)
versus α1β3γ2 GABA/Alprazolam-bound (α-red/β-blue/γ-gold; PDB 6HUO).
The impact of GABA/Alprazolam binding versus the antagonist is the same as
observed for the αβ receptor GABA binding versus antagonist (shown in **a**).
NOTE: ligands are not shown.

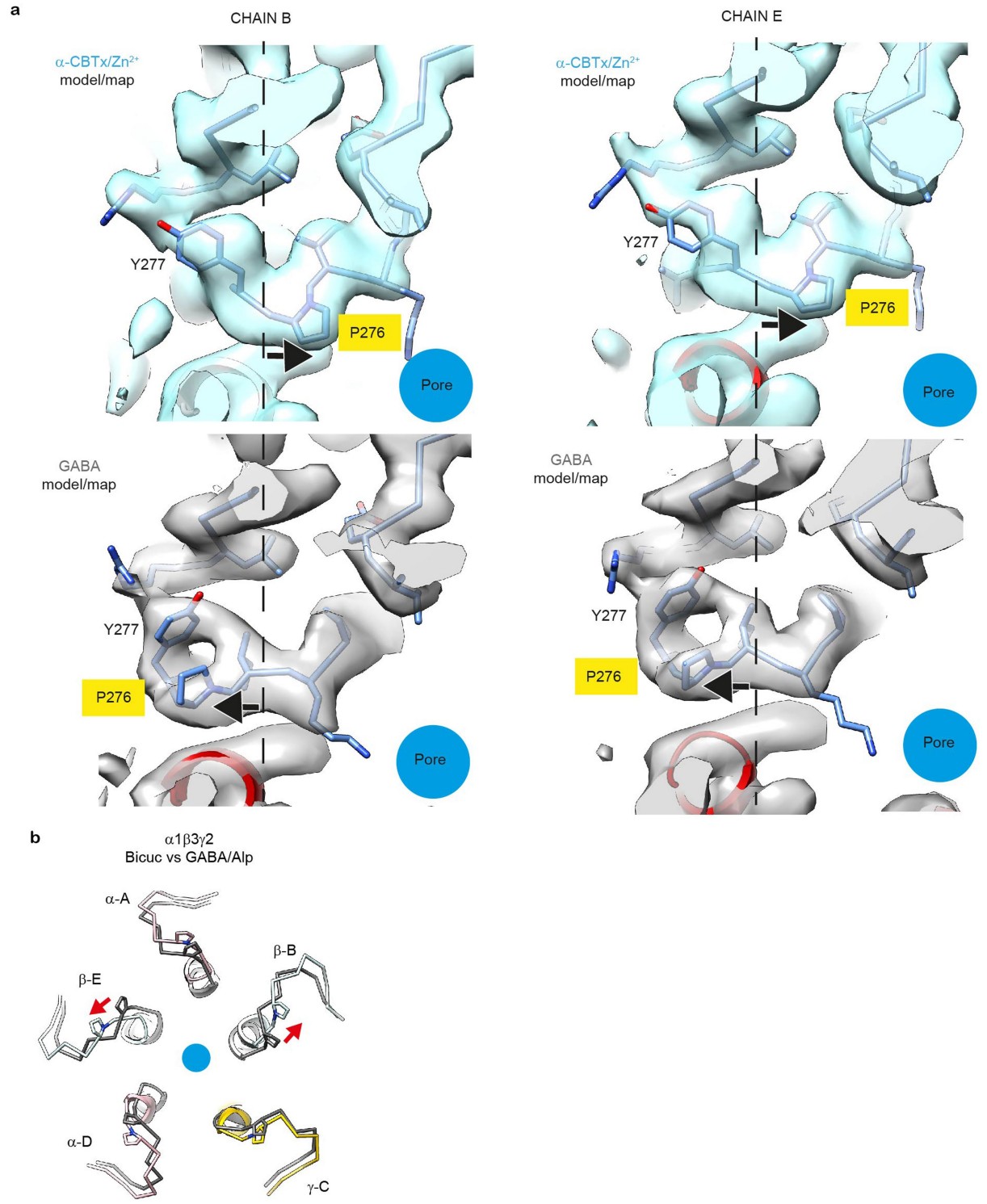

**Extended Data Fig. 7 | TMD M2-3 loop conformations. a,** Atomic model fits in the cryo-EM map density for the M2-M3 loops of β3 chains B and E for α-CBTx/Zn²⁺ (3.0 Å, blue maps) and GABA-bound (3.04 Å, grey maps) respectively. Viewed looking down on to the M2-M3 loop reveals the switch to the 'outward' conformation in response to the ECD binding GABA, as highlighted by Pro276 repositioning to the other side of the dashed line. **b,** Top-down view of α1β3γ2 bicuculline-bound (dark grey; PDB 6HUK) versus α1β3γ2 GABA/Alprazolam-bound (α-pink/β-pale blue/γ-gold; PDB 6HUO) showing the M2-M3 loop positions. In response to GABA binding the β-subunit chain B/E M2-M3 loops switch to the 'outward' conformation, indicated by red arrows that highlight the motion of Pro273. The γ2-subunit is in the outward conformation in both states.

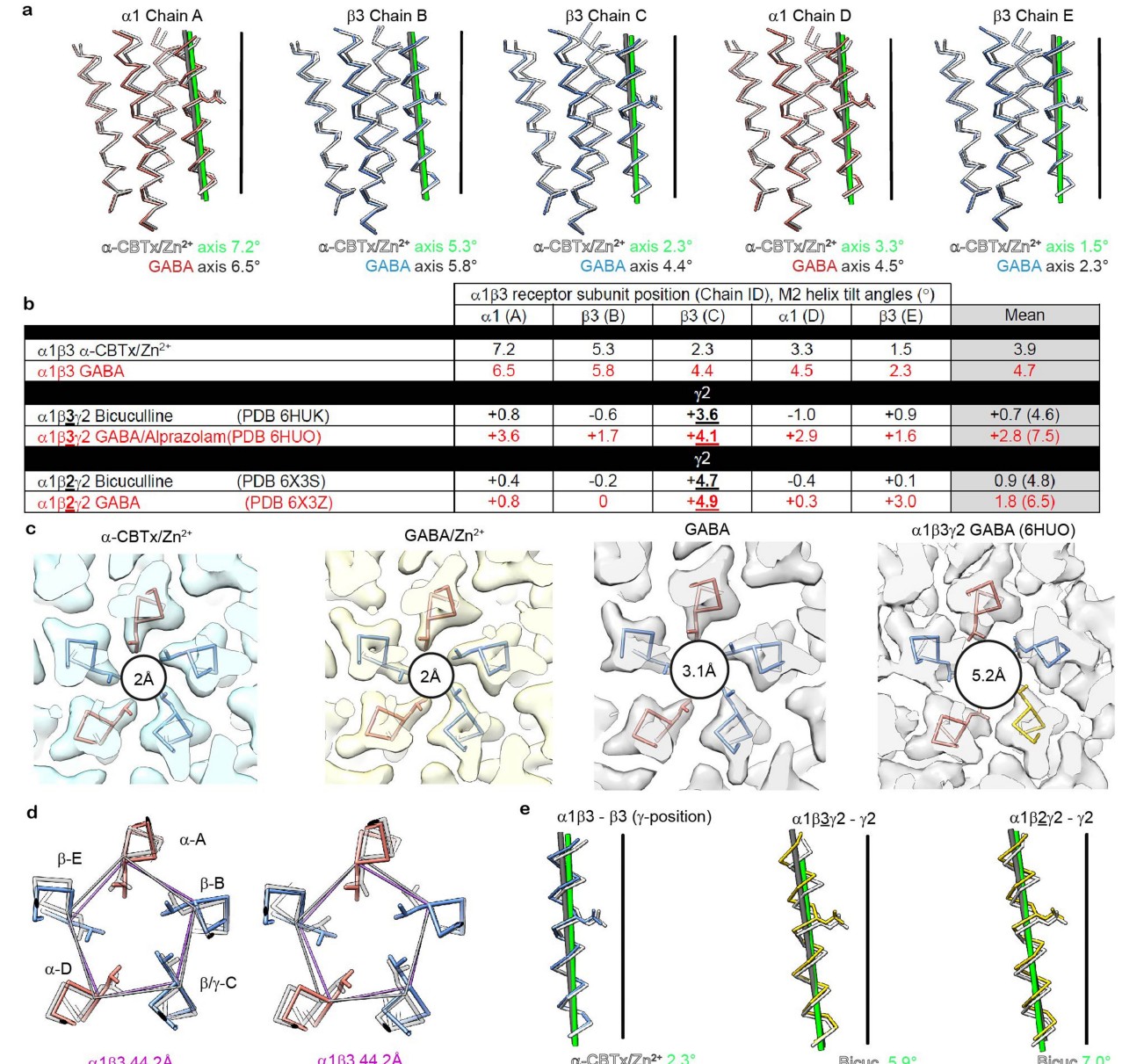

**Extended Data Fig. 8 | Pore arrangement. a**, Side-on views of subunit transmembrane helical bundle $C_\alpha$-polypeptide for Chains A-E showing M2 helix tilt axis. α1β3 GABA model is coloured red (α-subunit) or blue (β-subunit), and the M2 helix axis is shown as a dim grey bar. α1β3 α-CBTx/Zn²⁺ model is white, M2 helix axis green. Pore axis is to right of each bundle, vertical black bar. M2 helix tilt and/or translation away from pore axis increases for each subunit when GABA is bound (angle values shown; translations not measured but visible by eye). **b**, Table showing M2 helix tilt angles for αβ and αβγ receptors in antagonist (black text) and agonist (GABA) bound (red text) conformations. αβγ M2 helix tilt angles shown as increase (+) or decrease (−) relative to the equivalent αβ M2 helix. The biggest increase is for the γ2 subunits (values bold, underlined). **c**, Electron density map slices of the pore conformation at the 9′ Leu gate (pore diameters given inside pore circles) for α1β3 α-CBTx/Zn²⁺, GABA/Zn²⁺ and GABA-bound structures. For comparison the cryo-EM map of EMD-0282 used to build 6HUO PDB of GABA+Alprazolam bound structure is shown. **d**, Cross-section at 9′ Leu hydrophobic gate showing $C_\alpha$ pentagonal perimeters for GABA-bound αβ receptor versus GABA+Alprazolam-bound α1β**3**γ2 (left panel) or GABA-bound α1β**2**γ2 (right panel). **e**, Side by side comparison of the αβ receptor β3 subunit Chain C M2 helix tilts versus equivalent αβγ receptor γ2 M2 helix tilts, which are more reclined (for PDB codes see table in, **b**).

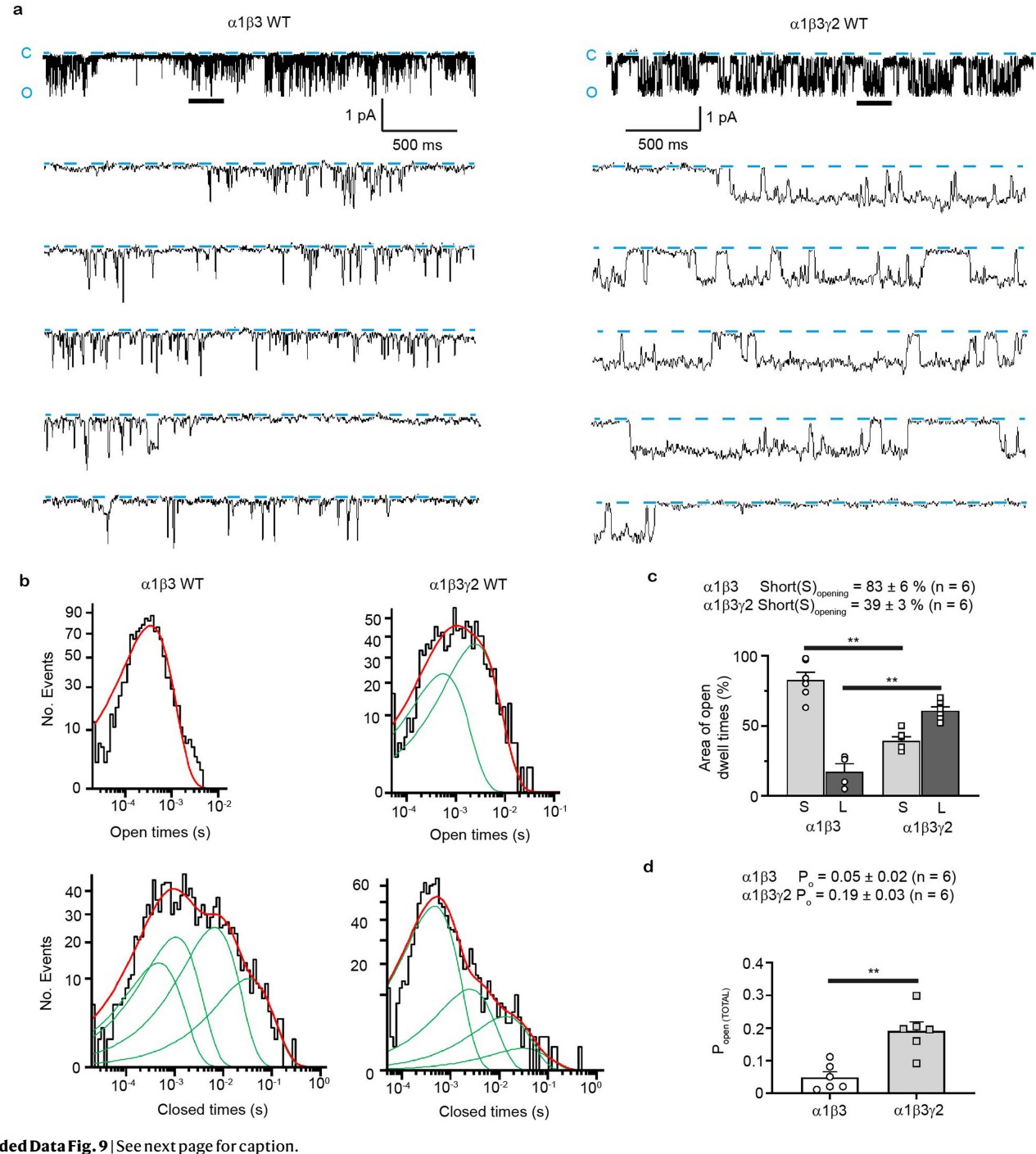

**Extended Data Fig. 9** | See next page for caption.

**Extended Data Fig. 9 | Single channel current analysis for GABA$_A$R heteromers. a**, epochs of GABA single channel currents recorded from outside-out patches of HEK293 cells expressing α1β3$_{WT}$ and α1β3γ2$_{WT}$ receptors, activated by 30 and 100 μM GABA respectively (-EC$_{95}$ for each receptor isoform) at low (upper trace) and higher time resolution (lower traces). C – closed and O – open state; closed state marked by dashed line; downward deflections are transitions to open state. **b**, Examples of open and closed state dwell time distributions for single cells expressing α1β3$_{WT}$ or α1β2γ2$_{WT}$ receptors. Single exponential component fits (green lines) and summed fits from a mixture of exponentials (red lines) are shown. In the example shown for open times, a single exponential fit was sufficient to account for the α1β3$_{WT}$ open state distribution, whilst for α1β3γ2$_{WT}$ a mixture of two exponentials was required. Mean exponential τ values (with SEM, and percentage area) determined from analysing multiple patches are: τ1 = 0.65 ± 0.15 ms (A1 = 83 ± 6%, n = 6), τ2 = 4.3 ± 1.1 ms (A2 = 17 ± 6%, n = 4 – two cells did not show long open times); for α1β3γ2$_{WT}$: τ1 = 0.78 ± 0.07 ms (A1 = 39 ± 3%, n = 6), τ2 = 4.8 ± 0.7 ms (A2 = 61 ± 3%, n = 6). For closed state dwell time distributions both receptor isoforms required a mixture of four exponentials of similar magnitudes, however α1β3 favoured the longer duration closed states, whereas α1β3γ2 favoured the shortest closed states which normally appear within bursts of openings. Mean τ values (and SEM, including percentage areas) from multiple patches are: α1β3$_{WT}$,: τ1 = 0.17 ± 0.05 ms (A1 = 36 ± 1%, n = 3), τ2 = 2.8 ± 0.3 ms (A2 = 62 ± 10%, n = 6), τ3 = 21 ± 5 ms (A3 = 22 ± 6%, n = 4), τ4 = 52 ± 3 ms (A4 = 20 ± 2%, n = 3)); α1β3γ2$_{WT}$: τ1 = 0.50 ± 0.04 ms (A1 = 68 ± 3%, n = 6), τ2 = 3.0 ± 0.3 ms (A2 = 21 ± 2%, n = 6), τ3 = 24 ± 4 ms (A3 = 9 ± 2%, n = 6), τ4 = 165 ± 25 ms (A4 = 2 ± 0.5%, n = 3). **c**, Bar graph showing percentage distribution between short versus long open state dwell times for α1β3 and α1β3γ2. Points represent mean ± s.e.m. n = 6, except for α1β3 long openings n = 4 (no long openings observed for two of the cells). Two-sided unpaired t-test comparisons of open dwell times t(8) = 7.53, p < 0.0001, and shut dwell times t(10) = 7.01, p < 0.0001. **d**, Bar graph showing the open probability (P$_o$; the average fraction of time spent in the open state), measured as the total open time divided by the total length of the recording. Recordings were taken from patches showing limited or no channel stacking (see Methods). Individual values are shown as circles/squares with associated error bars (mean ± s.e.m.), n = 6, two-sided unpaired t-test comparison, t(10) = 4.43, P = 0.0013. ** signifies as statistically different (P < 0.01) for bars linked by black lines. Each n = 1 value of an open time, shut time or P$_o$ were from biologically independent patch-clamp experiments from individual cells.

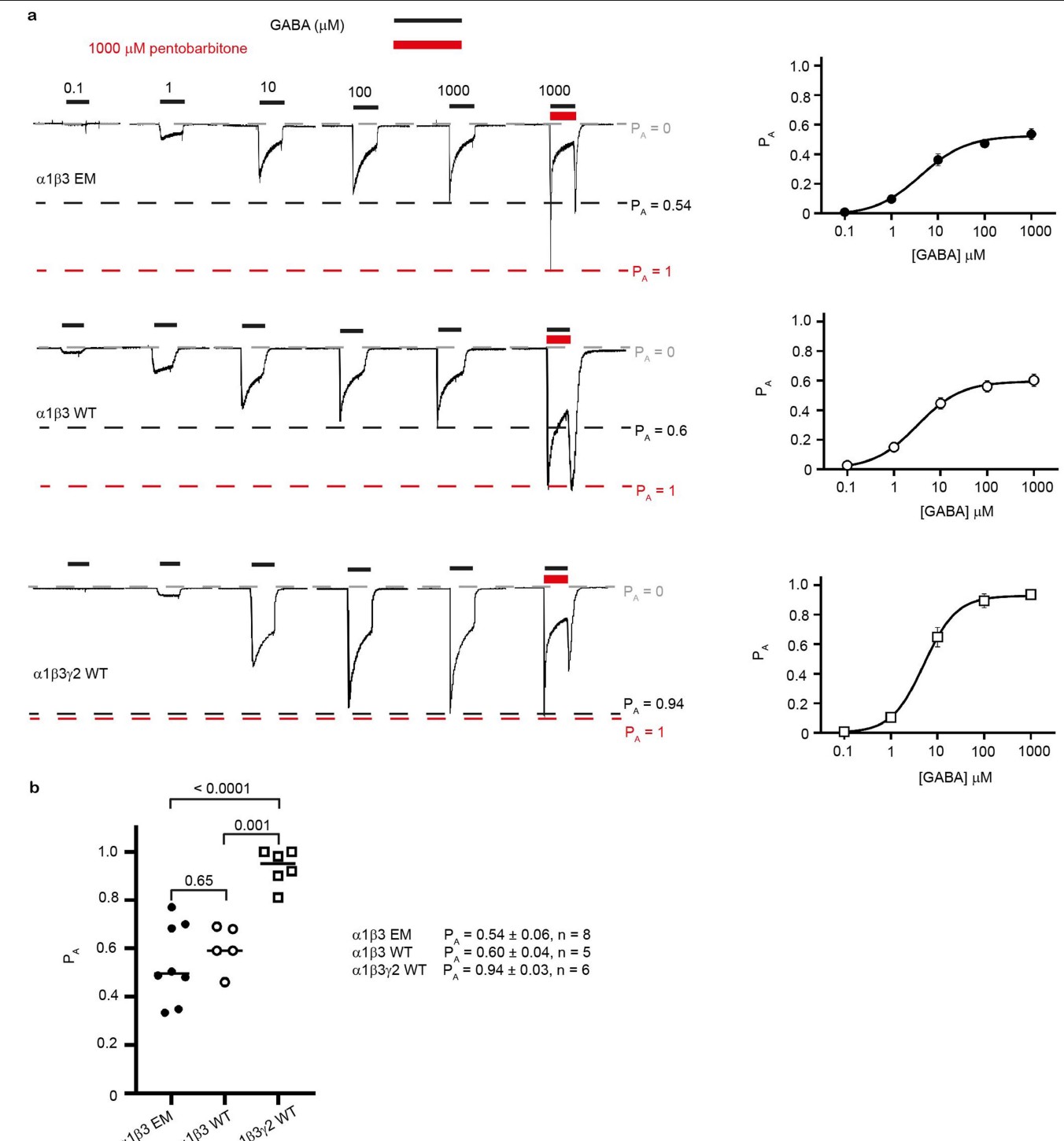

**Extended Data Fig. 10 | Probability of activation for α1β3 and α1β3γ2 receptors. a**, whole-cell patch clamp recordings from HEK293 cells expressing either α1β3 EM, α1β3 WT, or α1β3γ2 WT, showing responses to increasing concentrations of GABA and saturating GABA + 1 mM pentobarbitone in order to measure the probability of activation ($P_A$). Dashed lines indicate the baseline current (grey), the maximum activation by saturating GABA alone (black) and activation by saturating GABA + 1 mM pentobarbitone (red). Accompanying concentration response curve plots are provided (mean ± s.e.m, α1β3EM $n = 8$, α1β3WT $n = 5$, α1β3γ2 WT $n = 6$). **b**, Bar chart showing similar $P_A$ values for α1β3 EM and α1β3 WT, which were lower than for α1β3γ2 WT. Individual values are shown as circles/squares and bars are means ± s.e.m. (α1β3EM $n = 8$, α1β3WT $n = 5$, α1β3γ2 WT $n = 6$). One-way ANOVA comparing abWT vs abCryoEM vs abgWT: F(2,16) = 18.78; P < 0.0001. Post-hoc Tukey. Each $n = 1$ value of a $P_A$ were from biologically independent patch-clamp experiments from individual cells.

**Extended Data Table 1 | Cryo-EM data collection, refinement and validation statistics for GABA, GABA/Zn²⁺, α-CBTx/Zn²⁺**

| | α-CBTx/Zn²⁺ (EMDB-13315) (PDB-7PC0) | GABA/Zn²⁺ (EMDB-13314) (PDB-7PBZ) | GABA (EMDB-13290) (PDB-7PBD) |
|---|---|---|---|
| **Data collection and processing** | | | |
| Detector | Gatan K3 | Gatan K3 | Gatan K2 |
| Magnification | 105k | 130k | 130k |
| Energy filter slit width (eV) | 20 | 20 | 20 |
| Voltage (kV) | 300 | 300 | 300 |
| Flux on detector (e/pix/sec) | 18.3 | 15.1 | 5.94 |
| Electron exposure on sample (e–/Å²) | 50.44 | 53.27 | 53.9 |
| Target defocus range (μm) | 0.6-1.8 | 1.0-2.5 | 1.3-2.5 |
| Calibrated pixel size (Å) | 0.83 | 0.652 | 1.05 |
| Symmetry imposed | C1 | C1 | C1 |
| Extraction box size (pixels) | 400 | 520 | 320 |
| Initial particle images (no.) | 222901 | 434955 | 402052 |
| Final particle images (no.) | 78662 | 92749 | 139537 |
| Map resolution at FSC=0.143 (Å)* | 3.00 | 2.79 | 3.04 |
| Map resolution range | 2.7-11.7 | 2.5-5.6 | 2.7-11.7 |
| **Refinement** | | | |
| Initial model used (PDB) code | 7PBD+1YI5 Chain J | 7PBD | 6HUP |
| Model resolution at FSC=0.5 (Å) | 3.44 | 3.16 | 3.49 |
| Model composition | | | |
| Non-hydrogen atoms | 15695 | 14838 | 14850 |
| Protein residues | 1911 | 1774 | 1786 |
| Ligands | 21 | 37 | 26 |
| B factor (Å²) | | | |
| Protein | 120.60 | 106.11 | 125.98 |
| Ligand | 125.02 | 113.15 | 127.16 |
| R.m.s deviations | | | |
| Bond lengths (Å) | 0.004 | 0.004 | 0.004 |
| Bond angles (°) | 0.754 | 0.706 | 0.685 |
| Validation | | | |
| Molprobity score | 1.57 | 1.36 | 1.41 |
| Clashscore | 4.68 | 4.18 | 3.82 |
| Poor rotamers (%) | 0.06 | 0.0 | 0.0 |
| Ramachandran plot | | | |
| Favored (%) | 95.23 | 97.09 | 96.38 |
| Allowed (%) | 4.72 | 2.91 | 3.62 |
| Disallowed (%) | 0.05 | 0.0 | 0.0 |

# nature research

Trevor G. Smart

# Reporting Summary

Nature Research wishes to improve the reproducibility of the work that we publish. This form provides structure for consistency and transparency in reporting. For further information on Nature Research policies, see our Editorial Policies and the Editorial Policy Checklist.

## Statistics

For all statistical analyses, confirm that the following items are present in the figure legend, table legend, main text, or Methods section.

| n/a | Confirmed | |
|---|---|---|
| ☐ | ☒ | The exact sample size (*n*) for each experimental group/condition, given as a discrete number and unit of measurement |
| ☐ | ☒ | A statement on whether measurements were taken from distinct samples or whether the same sample was measured repeatedly |
| ☐ | ☒ | The statistical test(s) used AND whether they are one- or two-sided *Only common tests should be described solely by name; describe more complex techniques in the Methods section.* |
| ☒ | ☐ | A description of all covariates tested |
| ☒ | ☐ | A description of any assumptions or corrections, such as tests of normality and adjustment for multiple comparisons |
| ☐ | ☒ | A full description of the statistical parameters including central tendency (e.g. means) or other basic estimates (e.g. regression coefficient) AND variation (e.g. standard deviation) or associated estimates of uncertainty (e.g. confidence intervals) |
| ☐ | ☒ | For null hypothesis testing, the test statistic (e.g. *F*, *t*, *r*) with confidence intervals, effect sizes, degrees of freedom and *P* value noted *Give P values as exact values whenever suitable.* |
| ☒ | ☐ | For Bayesian analysis, information on the choice of priors and Markov chain Monte Carlo settings |
| ☒ | ☐ | For hierarchical and complex designs, identification of the appropriate level for tests and full reporting of outcomes |
| ☒ | ☐ | Estimates of effect sizes (e.g. Cohen's *d*, Pearson's *r*), indicating how they were calculated |

*Our web collection on statistics for biologists contains articles on many of the points above.*

## Software and code

Policy information about availability of computer code

| Data collection | Clampex 10.2, FEI EPU |
|---|---|
| Data analysis | Warp 1.09, cryoSPARC v3.2.0, UCSF Chimera v1.13.1, Pymol v2.1.1, Coot 0.9.3, Phenix 1.19.1, MolProbity, HOLE, Clampfit 10.2, Graphpad Prism 9, Origin 6, WinEDR |

For manuscripts utilizing custom algorithms or software that are central to the research but not yet described in published literature, software must be made available to editors and reviewers. We strongly encourage code deposition in a community repository (e.g. GitHub). See the Nature Research guidelines for submitting code & software for further information.

## Data

Policy information about availability of data

All manuscripts must include a data availability statement. This statement should provide the following information, where applicable:
- Accession codes, unique identifiers, or web links for publicly available datasets
- A list of figures that have associated raw data
- A description of any restrictions on data availability

Atomic model coordinates for alpha-CBTx/Zn2+, GABA/Zn2+ and GABA-bound structures have been deposited in the Protein Data Bank with accession codes 7PC0, 7PBZ, 7PBD, respectively. Cryo-EM density maps have been deposited in the Electron Microscopy Data Bank with accession codes EMD-13315, EMD-13314, EMD-13290 respectively.

# Field-specific reporting

Please select the one below that is the best fit for your research. If you are not sure, read the appropriate sections before making your selection.

☒ Life sciences  ☐ Behavioural & social sciences  ☐ Ecological, evolutionary & environmental sciences

For a reference copy of the document with all sections, see nature.com/documents/nr-reporting-summary-flat.pdf

# Life sciences study design

All studies must disclose on these points even when the disclosure is negative.

| | |
|---|---|
| Sample size | No statistical methods were used to estimate appropriate sample size. |
| Data exclusions | Following standard cryoSPARC processing pathways, best representative 2D and 3D classes were selected for the final particle reconstructions. Poor quality and irrelevant other classes were discarded. |
| Replication | Attempts to replicate/reproduce the data were successful as detailed in the electrophysiology replicates. For cryo-EM Two independent maps of each cryo-EM sample were generated in order to estimate resolution according to the recommended procedures in the field (the 'gold standard'). |
| Randomization | EM particle sets were randomly split for the purposes of estimating overall resolution. Otherwise randomization was not relevant to these studies. |
| Blinding | Blinding was not relevant to this study. |

# Reporting for specific materials, systems and methods

We require information from authors about some types of materials, experimental systems and methods used in many studies. Here, indicate whether each material, system or method listed is relevant to your study. If you are not sure if a list item applies to your research, read the appropriate section before selecting a response.

### Materials & experimental systems

| n/a | Involved in the study |
|---|---|
| ☐ | ☒ Antibodies |
| ☐ | ☒ Eukaryotic cell lines |
| ☒ | ☐ Palaeontology and archaeology |
| ☒ | ☐ Animals and other organisms |
| ☒ | ☐ Human research participants |
| ☒ | ☐ Clinical data |
| ☒ | ☐ Dual use research of concern |

### Methods

| n/a | Involved in the study |
|---|---|
| ☒ | ☐ ChIP-seq |
| ☒ | ☐ Flow cytometry |
| ☒ | ☐ MRI-based neuroimaging |

## Antibodies

| | |
|---|---|
| Antibodies used | Rho-1D4 antibody was purchased from the University of British Columbia. The megabody Mb25 was obtained from the laboratory of Professor Jan Steyeart (VUB). Nanobody Nb25 was made in the lab. |
| Validation | Rho-1D4 validated by ability to purify GABA-A-R. The nanobody Nb25, used to design the megabody Mb25 as described in methods, was characterised and published elsewhere (PMID: 28991263). Mb25 has been characterised and published elsewhere (PMID: 33408403). |

## Eukaryotic cell lines

Policy information about cell lines

| | |
|---|---|
| Cell line source(s) | HEK 293S GnTI- cells were obtained from ATCC. |
| Authentication | Further authentication was not performed for this study. |
| Mycoplasma contamination | Mycoplasma testing was not performed for this study. |
| Commonly misidentified lines (See ICLAC register) | No commonly misidentified cell lines were used in this study. |

