## [Peer Review File · Nature]

Manuscript Title: Mechanisms of inhibition and activation of extrasynaptic $\alpha\beta$ GABA_A receptors

Reviewer Comments & Author Rebuttals

Reviewer Reports on the Initial Version:

Referee #1 (Remarks to the Author):

This article by Kasaragod et al., reported three cryoEM structures of the extrasynaptic $\alpha\beta$ GABA receptors in CBTx/Zn, GABA/Zn, or GABA-bound state. The structures were determined at 2.8-3 Å, which are sufficient for the structural interpretation described in the article. These structures elucidate the inhibition mechanism of CBTx and Zn. The Zn results are particularly interesting and show how divalent cations block an anion channel. Furthermore, the GABA-bound structure reveals a different orientation of the M2 helices, which results in a narrow pore in the $\alpha\beta$ GABA receptor compared to the $\alpha\beta\gamma$ receptor. This observation provides a possible structural explanation why the $\alpha\beta$ GABA receptor has a lower P_o . This work is a good addition to the field of GABA receptors. Given the importance of the extrasynaptic GABA receptor in neuroscience, I would recommend its publication after the following points are addressed or discussed.

1. Could the authors further elaborate on the biological significance of CBTx binding on GABARs? The peripheral nACh receptors are targeted by snake toxins. Will GABA receptors ever encounter snake toxins in a biological setting?
2. Are there any references or data showing that Mb25 does not allosterically affect the binding/inhibition of CBTx? It would be nice to perform some functional experiments to make sure this.
3. Line 17. The authors described the CBTx/Zn structure as "resting state". To me a resting state is like an apo state and the CBTx/Zn structure is more like an inhibited state.
4. Line 71. "No GABA_A receptor-toxin structures have been elucidated." Please revise this sentence to be more precise because there are picrotoxin structures.
5. Line 100. " β -subunit chains B/E=0.9 Å" but in extended fig 3l the RMSD is stated as 0.8 Å.
6. Fig 2a. The y-axis title already says that this is inhibition. It is thus a bit confusing that the numbers have minus signs.
7. Line 196-197. "..... increase the pore diameter from 2 Å to 3.1 Å." How do the authors get these numbers? In Fig3 c,d, the diameter at 9' ring of GABA structure is 1.6 Å, not 3.1 Å. Along the same line, in extended fig 7c the diameters shown are different from the HOLE calculation. Please state how these numbers are derived.
8. Line 202. "...from 4.7° to 7.5°." In extended fig 7b, the number is 4.6, not 4.7.
9. Line 213. "The smaller 9' gate observed here for the $\alpha 1\beta 3$ receptor cannot be explained on methodological grounds because the same nanodisc sample preparation was used for the $\alpha 1\beta 3\gamma 2$ receptor reported with open 9' gate". This is a reasonable inference but should not be stated as a fact. There is no reason why things work for the $\alpha\beta\gamma$ type will definitely work for the $\alpha\beta$ type. For example, the opening of the $\alpha\beta$ type may require a certain lipid composition/ratio that is not recapitulated by the POPC:BBL lipid mixture.
10. Extended fig 1. GABA/Zn dataset. Are those resolution numbers next to the FSC labels correct? Looks like they are just copied from the CBTx/Zn dataset. At least for the GABA/Zn one, the "Corrected" resolution should be 2.8 Å, not 3 Å.
11. Extended fig 1. Please state what program is used to calculate local resolution.
12. Extended fig 7a. Here CBTx/Zn dataset is labeled as CBTx which is fine, but in Extended fig 7e the same dataset is labeled as CBTx/Zn. Please be consistent to avoid confusion.

Referee #2 (Remarks to the Author):

Kasaragod et al. report cryo-EM structures of $\alpha 1\beta 3$ receptors in complex with combinations of GABA/ Zn²⁺/ α -CBTx to elucidate the structure and mechanics of Zn²⁺ modulation and the binding site and binding mode of the toxin. An allosterically inhibiting Zn²⁺ site was observed in the uppermost part of the pore, a finding fully consistent with mutational evidence and of high interest. In addition, the authors observe a different pore diameter at the 9' gate level, which they interpret to lead to a lower open probability (Po) in response to GABA. The structural biology is further supported by some functional data. The work is of interest, the presentation of the results is clear and technically very good – however, there are some points in the introduction as well as in the interpretation of the findings that require at least some clarification as indicated point by point below.

General point(s):

In the title and in 17, “ $\alpha\beta$ receptors” are advertised – the study presents $\alpha 1\beta 3$ receptors. This should be specified, for reasons detailed below in individual comments.

The simple grouping into synaptic and extrasynaptic receptors falls short of acknowledging the fact that some populations are found both in synaptic and non-synaptic localizations and suggests simple groups of properties common to all “extrasynaptic” receptors – this reviewer is very skeptical about such simple models.

Line 23 “key novel traits adopted by extrasynaptic receptors” is misleading in several ways, and should be reworded. In evolution, somatic receptors existed before synapses – so which is “novel”? Not all receptors found in neuronal non-synaptic compartments exhibit the same properties, e.g. $\alpha 5\beta 2$ do not share the high Zn²⁺ sensitivity that is of interest here.

Line 43. It is suggested that all “extrasynaptic receptors” have low Po upon GABA binding. In recombinant expression systems, many laboratories have observed high Po for $\alpha 1\beta 2$, $\alpha 1\beta 3$ or even $\alpha 1\beta 1$. Since this is central to the interpretation of the structures, data is needed that shows for a side by side comparison that $\alpha 1\beta 3$ has a lower Po than $\alpha 1\beta 3\gamma 2$, which is what is claimed here.

Lines 28-30: To introduce the pLGIC superfamily, the authors mention mammalian receptors and bacterial homologues – quite a jump in evolution, without acknowledging the rich diversity of pLGICs found in invertebrates. A more balanced set of examples, or explicit limitation to mammalian family members would be preferred.

Line 33: “extrasynaptic receptors linked ---- to asthma” - ??? Targeting GABA-A receptors in airway smooth muscle and epithelium cannot be attributed to “extrasynaptic” receptors, as smooth muscle and epithelial cells do not have synapses. The authors should explicitly indicate that GABA-A receptors are found in many non-neuronal and non-excitabile cells where they contribute to cell functions different from “tonic inhibition”.

Interpretation: The structural facts stand as they are, but the interpretation of the narrower 9' gate in terms of receptor dynamics seems speculative, and is not supported by evidence from e.g. single channel recordings and/ or MD. I am not suggesting to do MDs (without intracellular domain of limited value anyhow), I simply suggest to tone down the arguments and claims on the low Po for the $\alpha 1\beta 3$ receptors (for $\alpha 4\beta$ or $\alpha 6\beta$ literature data seems very supportive of such low Po, but not $\alpha 1\beta$).

Overall, after careful rewriting, this reviewer is supportive of publishing this work. margot ernst

Author Rebuttals to Initial Comments:

Referee #1 (Remarks to the Author):

This article by Kasaragod et al., reported three cryoEM structures of the extrasynaptic $\alpha\beta$ GABA receptors in CBTx/Zn, GABA/Zn, or GABA-bound state. The structures were determined at 2.8-3 Å, which are sufficient for the structural interpretation described in the article. These structures elucidate the inhibition mechanism of CBTx and Zn. The Zn results are particularly interesting and show how divalent cations block an anion channel. Furthermore, the GABA-bound structure reveals a different orientation of the M2 helices, which results in a narrow pore in the $\alpha\beta$ GABA receptor compared to the $\alpha\beta\gamma$ receptor. This observation provides a possible structural explanation why the

$\alpha\beta$ GABA receptor has a lower P_o . This work is a good addition to the field of GABA receptors. Given the importance of the extrasynaptic GABA receptor in neuroscience, I would recommend its publication after the following points are addressed or discussed.

Thank you very much for reviewing our manuscript, for your feedback, and for taking the time to carefully check specific data values and spot mistakes. This is very much appreciated.

1. Could the authors further elaborate on the biological significance of CBTx binding on GABARs? The peripheral nACh receptors are targeted by snake toxins. Will GABA receptors ever encounter snake toxins in a biological setting?

Interesting point. We cannot say with certainty. Although the sensitivity of GABA-A receptors to alpha-cobratoxin is reasonable, being sub-micromolar, and a snake-bite of a small rodent might lead to injection of sufficient toxin to theoretically bind to GABA-A receptors, it will have the challenge of permeating through the blood brain barrier to reach CNS GABA-A receptors. Thus, we suggest it is unlikely to reach GABA-A receptors in a physiologically meaningful way.

In the first paragraph after the “a-Cobratoxin mechanism of inhibition” section we have now clarified any possible biological significance of CBTx in the paper by changing the original sentence from: “ α -CBTx blocks muscle nAChRs to paralyse prey, but it is also an effective ‘three-finger’ inhibitor of most types of GABA_A receptor”

To:

“ α -CBTx blocks muscle nAChRs to paralyse prey, but more recently has been shown to act with reduced potency as an inhibitor of GABA_A receptors in recombinant expression systems”

Despite the physiological considerations, the alpha-cobratoxin structure was important to give us an inhibited conformation for comparison to the GABA bound state. Also, as originally stated in the text, toxins can serve as new scaffolds for the design of subtype selective inhibitors, which are not currently available for GABA-A receptors (with the exception of alpha 5-receptors). The toxin-bound structure can guide engineering approaches in the future.

2. Are there any references or data showing that Mb25 does not allosterically affect the binding/inhibition of CBTx? It would be nice to perform some functional experiments to make sure this.

We appreciate this and have carefully considered this point. However, we cannot conceive a clear/clean experiment that would give an unequivocal result in the context of the paper and for the structures that we have solved.

In the ‘inhibited’ receptor structure we show the extracellular domain of the $\alpha 1\beta 3$ GABA-A receptor is bound by saturating levels of CBTx and Mb25. The receptor’s extracellular domain adopts the same conformation to that previously presented as an inhibited state for $\alpha 1\beta 3\gamma 2$ (2018 Masiulis S, Aricescu AR, PMID: 30602790). Thus, the inhibitor CBTx is bound to a conformation that is completely consistent with an inhibited state. Therefore, at saturating CBTx, Mb25 is having no impact on CBTx’s ability to stabilise the inhibited state in our structure. Thus, if we were to perform a functional experiment with saturating CBTx plus Mb25, we would simply see complete inhibition, which is the functional outcome we observe from exposure to CBTx alone, and also shown in our ‘inhibited’ structure for CBTx+Mb25. Even if we instead performed experiments at non-saturating CBTx concentrations, and observed that Mb25 had some impact on non-saturating doses of CBTx, any interpretation would be unclear as our structure is bound by saturating CBTx (both binding sites

are occupied and a saturating dose of 10 μM CBTx was used in cryo-EM). Of course, we can do the experiment but we are doubtful it will show anything of significance due to the above considerations.

As a footnote, the CBTx is extremely expensive, at $\sim\text{£}400$ for 100 μg (CBT001 from Smartox), which gives only 12 ml at the lowest saturating dose of 1 μM , and would mean we would need, for the electrophysiology experiments, at least $\text{£}1000$ of toxin, and possibly considerably more. Whilst we are happy to spend such a sum on necessary experiments, we feel that in this instance it is probably unnecessary with little gain to be had in our understanding.

3. Line 17. The authors described the CBTx/Zn structure as “resting state”. To me a resting state is like an apo state and the CBTx/Zn structure is more like an inhibited state.

A good point. We have now replaced “resting” with “inhibited”, and also corrected this at other points in the text where appropriate.

4. Line 71. “No GABA_A receptor-toxin structures have been elucidated.” Please revise this sentence to be more precise because there are picrotoxin structures.

Agreed. We have now changed this to “...selective inhibitor design but no GABA_A receptor structures in complex with protein inhibitors have been elucidated...”

5. Line 100. “ β -subunit chains B/E=0.9 Å” but in extended fig 3I the RMSD is stated as 0.8 Å.

Thanks, this is now corrected in main text: “ β -subunit chains B/E=0.8 Å”

6. Fig 2a. The y-axis title already says that this is inhibition. It is thus a bit confusing that the numbers have minus signs.

We agree. “Inhibition” has been replaced by “modulation”.

7. Line 196-197. “..... increase the pore diameter from 2 Å to 3.1 Å.” How do the authors get these numbers? In Fig3 c,d, the diameter at 9' ring of GABA structure is 1.6 Å, not 3.1 Å. Along the same line, in extended fig 7c the diameters shown are different from the HOLE calculation. Please state how these numbers are derived.

We apologise for the confusion and are grateful this mistake has been spotted. The confusion is simply caused by a mistake on our part in using the term “diameter” in Fig.3 when we should have used “radius”. So for Fig.3c,d a radius of 1.6 Å (actually 1.55 Å) correlates with the text mention of a diameter of 3.1Å. We have now corrected Fig.3c,d to change the diameter term and symbols to radius. In this regard, EDF.7c, which shows the same diameter values as the text, is now consistent with the radius values in Fig.3c,d.

8. Line 202. “...from 4.7° to 7.5°.” In extended fig 7b, the number is 4.6, not 4.7.

Thanks, this is now corrected in the main text to 4.6°

9. Line 213. “The smaller 9' gate observed here for the $\alpha 1\beta 3$ receptor cannot be explained on methodological grounds because the same nanodisc sample preparation was used for the $\alpha 1\beta 3\gamma 2$ receptor reported with open 9' gate”. This is a reasonable inference but should not be stated as a fact. There is no reason why things work for the $\alpha\beta\gamma$ type will definitely work for the $\alpha\beta$ type. For

example, the opening of the $\alpha\beta$ type may require a certain lipid composition/ratio that is not recapitulated by the POPC:BBL lipid mixture.

Yes we agree, and acknowledge that there are caveats and complexities. Due to the addition of new functional experiments and the limitations on word count, we have now removed this sentence.

10. Extended fig 1. GABA/Zn dataset. Are those resolution numbers next to the FSC labels correct? Looks like they are just copied from the CBTx/Zn dataset. At least for the GABA/Zn one, the “Corrected” resolution should be 2.8 Å, not 3 Å.

Yes, our apologies, the text block was copied and then the correct values were not inserted. This has now been corrected.

11. Extended fig 1. Please state what program is used to calculate local resolution.

We have updated the methods section under the section: Cryo-electron microscopy data acquisition and image processing, with the following information:

“A local_res map was generated in cryoSPARC using the program “local resolution estimation”. The resolution range was based on the FSC output calculated for voxels only within the mask output from the homogenous refinement job used as the input for local resolution estimation. To generate maps colored by local resolution, the local_res map along with the main map were opened in UCSF Chimera62 and processed using the surface color tool.”

In the EDF.1 figure legend we state “For the three structures, α -CBTx/Zn²⁺, GABA/Zn²⁺, and GABA, a map on the left is coloured by local resolution (see methods)”.

12. Extended fig 7a. Here CBTx/Zn dataset is labeled as CBTx which is fine, but in Extended fig 7e the same dataset is labeled as CBTx/Zn. Please be consistent to avoid confusion.

We acknowledge this confusion. Throughout the text, and all figures, we always refer to this structure as α -CBTx/Zn²⁺. However, for Fig. 7a we could not fit in the full name, and so purely for aesthetic reasons in this one instance we dropped the “Zn²⁺” label. However, given the confusion caused, we have now changed this label to drop the “ α -” instead, which allows us to label as follows “CBTx/Zn”, rather than the usual “ α -CBTx/Zn²⁺” which is too long.

Referee #2 (Remarks to the Author):

Kasaragod et al. report cryo-EM structures of $\alpha 1\beta 3$ receptors in complex with combinations of GABA/ Zn²⁺/ α -CBTx to elucidate the structure and mechanics of Zn²⁺ modulation and the binding site and binding mode of the toxin. An allosterically inhibiting Zn²⁺ site was observed in the uppermost part of the pore, a finding fully consistent with mutational evidence and of high interest. In addition, the authors observe a different pore diameter at the 9' gate level, which they interpret to lead to a lower open probability (Po) in response to GABA. The structural biology is further supported by some functional data. The work is of interest, the presentation of the results is clear and technically very good – however, there are some points in the introduction as well as in the interpretation of the findings that require at least some clarification as indicated point by point below.

We thank the referee for reviewing our manuscript and are very grateful for the insightful feedback.

General point(s):

In the title and in 17, “ab receptors” are advertised – the study presents $\alpha 1\beta 3$ receptors. This should be specified, for reasons detailed below in individual comments.

We appreciate this point. However, the Nature rules stipulate a title allowance of 75 characters including spaces and our current title “Mechanisms of inhibition and activation of extrasynaptic $\alpha\beta$ GABA_A receptors” takes 75 characters. Thus, to be more specific and replace $\alpha\beta$ with $\alpha 1\beta 3$ would go over the limit, and require changing the title, and we feel that overall this title represents the story of the paper best. If the allowance is flexible, we can change our title ($\alpha\beta$ to $\alpha 1\beta 3$).

Furthermore, we provide strong new evidence now included in the paper (see below) that $\alpha 1\beta 3$ do exhibit a lower P_o and so are representative of other $\alpha\beta$ extrasynaptic GABA-A-Rs such as $\alpha 4\beta 3$. Thus, we feel that the current title is probably a fair reflection of the story.

The simple grouping into synaptic and extrasynaptic receptors falls short of acknowledging the fact that some populations are found both in synaptic and non-synaptic localizations and suggests simple groups of properties common to all “extrasynaptic” receptors – this reviewer is very skeptical about such simple models.

We have now moderated the tone of the summary + intro paragraph to avoid concrete statements e.g.

On reflection, we completely accept the referee’s view and we have now revised the abstract and introduction to avoid implying an absolute distinction between extrasynaptic and synaptic subtypes. For example, the first sentence of the opening summary paragraph is: “Extrasynaptic gamma-aminobutyric acid (GABA) Type-A receptors, such as $\alpha\beta$, $\alpha 4/6\beta d$ and $\alpha 5\beta g$ receptors, mediate an essential persistent (tonic) inhibitory conductance in many regions of the mammalian brain”

This statement allows room for inclusion of other subtypes.

Another example, for the 4th sentence of the opening summary paragraph reads:

“Tonic GABAergic responses are tailored to avoid over-suppressing neuronal communication, and often exhibit high sensitivity to Zn^{2+} blockade, which contrasts with synapse preferring $\alpha 1/2/3\beta g$ receptor responses.”

Again, this statement allows room for the interpretation that some $\alpha 1/2/3\beta g$ receptors can be extrasynaptic instead.

Due to the word limit, it is not possible to provide much detail.

Line 23 “key novel traits adopted by extrasynaptic receptors” is misleading in several ways, and should be reworded. In evolution, somatic receptors existed before synapses – so which is “novel”? Not all receptors found in neuronal non-synaptic compartments exhibit the same properties, e.g. $\alpha 5\beta g 2$ do not share the high Zn^{2+} sensitivity that is of interest here.

We appreciate this point of view and accept the referee’s assertion. We have now moderated the abstract and introduction to reflect this. As an example:

“Overall, this study explains key novel traits adopted by extrasynaptic receptors to optimise them for extrasynaptic localisation and function”

Has been changed to:

“Overall, this study explains distinct traits adopted by alpha-beta receptors that adapt them to a role in tonic signalling”

Line 43. It is suggested that all “extrasynaptic receptors” have low P_o upon GABA binding. In recombinant expression systems, many laboratories have observed high P_o for $\alpha 1\beta 2$, $\alpha 1\beta 3$ or even $\alpha 1\beta 1$. Since this is central to the interpretation of the structures, data is needed that shows for a side by side comparison that $\alpha 1\beta 3$ has a lower P_o than $\alpha 1\beta 3\gamma 2$, which is what is claimed here.

We acknowledge that it would be very useful and informative to have data in the paper that directly informs on channel P_o . We are aware of a range of P_o values from others, including published data indicative of a lower P_o for $\alpha 1\beta 3$, for example from 2020 Akk G Steinbach JH PMID:32873746. Given this, we determined P_o under our conditions and now provide compelling new data indicating that $\alpha 1\beta 3$ has a lower P_o than its $\alpha 1\beta 3\gamma 2$ counterpart in side-by-side comparisons using both whole-cell recordings to measure a probability of activation (P_A) and by directly determining P_o in single channel recordings. These new data provide significant functional support towards our interpretation of the $\alpha\beta$ -receptor structures. Details are provided below -

SINGLE CHANNEL RECORDING EXPERIMENTS:

We provide new single channel recording data comparing $\alpha 1\beta 3$ and $\alpha 1\beta 3\gamma 2$ side-by-side when exposed to concentration-matched (near-saturating, EC_{95}) GABA. This revealed a clear and significant reduction in the open state dwell time of $\alpha 1\beta 3$ receptors versus $\alpha 1\beta 3\gamma 2$ receptors. For both receptors, open state distributions revealed one short and one long open state dwell time of similar durations. However, for $\alpha 1\beta 3$ receptors, ~83 % were brief openings and only ~17 % were long openings, whereas for $\alpha 1\beta 3\gamma 2$ receptors only ~39 % were brief openings, the majority being longer openings, ~61 %. Furthermore, whilst both receptors exhibited four closed state dwell times of similar magnitudes, $\alpha 1\beta 3$ favoured the longer duration closed states, whereas $\alpha 1\beta 3\gamma 2$ favoured the shortest closed states which normally appear within bursts of openings (shown in extended data Fig 10). Overall, the data is unequivocally consistent with $\alpha 1\beta 3$ receptors having a reduced P_o and also validates that $\alpha 1\beta 3$ tends to enter an open state briefly, which precludes its capture by cryo-EM, as observed in our structures.

For $\alpha 1\beta 3\gamma 2$, it is possible to accurately estimate P_o from single channel recordings by analysing the amount of time the channel spends in the open state during a burst (intra-burst P_o normally giving values around 0.8). However, because $\alpha 1\beta 3$ receptors only exhibit brief openings, and lack defined burst structure owing to their low P_o , it is not possible to measure a burst P_o , hence we supplemented the single channel recording with the approach of measuring P_A (see below – Whole-cell recording). Nevertheless, we assessed the open probabilities from continuous single channel recordings where there was no evidence of channel stacking and thus it was possible, but not guaranteed, that these patches contained only one active channel. Even if this premise is false, the same analysis conditions were applied to recordings for both $\alpha 1\beta 3\gamma 2_{WT}$ and $\alpha 1\beta 3_{WT}$ receptors. Despite this caveat, the analysis of open probability still clearly revealed that P_o is significantly higher for $\alpha 1\beta 3\gamma 2_{WT}$ than for $\alpha 1\beta 3_{WT}$ receptors indicating that gating of $\alpha 1\beta 3\gamma 2_{WT}$ ion channels is more efficient than for $\alpha 1\beta 3_{WT}$. Again, this finding supports the observation of a mostly closed channel in GABA-bound $\alpha 1\beta 3$ by cryo-EM, as we observed.

This data is discussed in the main text in the final paragraph of the section “receptor response to GABA” and presented in new EDF9.

WHOLE-CELL RECORDING EXPERIMENTS:

To do this we measured the term P_A , which is the probability of being in the active state, and has been used previously to observe whether or not an agonist P_o is close to 1 or not, i.e. an indicative proxy of relative P_o (2020 Akk G Steinbach JH, PMID:32873746). In brief, using whole-cell recordings this compares the maximal response achieved by a particular agonist, and this is compared to the maximum possible response that can be observed for that receptor attained by using saturating

agonist + a positive allosteric modulator (PAM, e.g., a barbiturate). For gamma containing receptors, which already have a near maximal P_o (close to 1) in response to saturating GABA, the GABA + PAM response cannot increase much further (as P_o cannot be greater than 1). However, for any receptor exhibiting a lower P_o the agonist + PAM response will be able to increase the P_o and so the response will be significantly larger. This is precisely what we observed in side-by-side comparisons of $\alpha 1\beta 3$ versus $\alpha 1\beta 3\gamma 2$ receptors, with the P_A for GABA versus GABA + pentobarbitone being $\sim 0.5-0.6$ for $\alpha 1\beta 3$ WT and the $\alpha 1\beta 3$ cryo-EM construct, very close to that previously reported for $\alpha 1\beta 3$ for muscimol versus muscimol + propofol (~ 0.6 - 2020 Akk G Steinbach JH, PMID:32873746). Whereas for $\alpha 1\beta 3\gamma 2$ we observed a P_A of >0.9 . The difference between the $\alpha\beta$ versus $\alpha\beta\gamma$ P_A was significant ($P < 0.001$).

This data is also discussed in the main text in the final paragraph of the section “receptor response to GABA” and presented in new EDF10.

Lines 28-30: To introduce the pLGIC superfamily, the authors mention mammalian receptors and bacterial homologues – quite a jump in evolution, without acknowledging the rich diversity of pLGICs found in invertebrates. A more balanced set of examples, or explicit limitation to mammalian family members would be preferred.

We appreciate this point and therefore text is revised to: “...belong to the pentameric ligand-gated ion channel (pLGIC) superfamily which includes mammalian nicotinic acetylcholine receptors (nAChRs), serotonin Type-3A (5HTA), and glycine (Gly) receptors, as well as other non-mammalian homologues”

Line 33: “extrasynaptic receptors linked ---- to asthma” - ??? Targeting GABA-A receptors in airway smooth muscle and epithelium cannot be attributed to “extrasynaptic” receptors, as smooth muscle and epithelial cells do not have synapses. The authors should explicitly indicate that GABA-A receptors are found in many non-neuronal and non-excitabile cells where they contribute to cell functions different from “tonic inhibition”.

Yes agreed. This is a good point, and for the sake of clarity and simplicity we now remove the reference to asthma.

Interpretation: The structural facts stand as they are, but the interpretation of the narrower 9' gate in terms of receptor dynamics seems speculative, and is not supported by evidence from e.g. single channel recordings and/ or MD. I am not suggesting to do MDs (without intracellular domain of limited value anyhow), I simply suggest to tone down the arguments and claims on the low P_o for the $\alpha 1\beta 3$ receptors (for $\alpha 4\beta$ or $\alpha 6\beta$ literature data seems very supportive of such low P_o , but not $\alpha 1\beta$).

We agree that previously the interpretation of the narrower 9' gate in terms of receptor dynamics was not adequately supported by accompanying functional data. However, as described above, we now provide compelling evidence that the $\alpha 1\beta 3$ receptor does actually have a lower P_o than its $\alpha 1\beta 3\gamma 2$ counter-part, both from whole-cell recordings using a measure, P_A , and directly from single channel recordings comparing open state dwell time distributions. In light of these new data, we feel that our interpretation of the narrower 9' gate being a reflection of a lower P_o is justified. It is logical and seemingly the best possible explanation for our structural observations. We have included new text to ensure this important point is made clear to the reader.

Overall, after careful rewriting, this reviewer is supportive of publishing this work. margot ernst

Reviewer Reports on the First Revision:

Referee #1 (Remarks to the Author):

The authors have addressed my questions to my satisfaction in their revised manuscript. They have performed additional analyses/experiments to support their conclusions. I have no further issues.

Referee #2 (Remarks to the Author):

The authors have added very convincing functional data and addressed all concerns, "good to go"